# miRNA-target complementarity in cnidarians resembles its counterpart in plants

Yael Admoni [1]✉, Arie Fridrich [1], Paris K Weavers [2], Reuven Aharoni[1], Talya Razin [3], Miguel Salinas-Saavedra [2], Michal Rabani [3], Uri Frank [2] & Yehu Moran [1]✉

## Abstract

**microRNAs (miRNAs) are important post-transcriptional regulators that activate silencing mechanisms by annealing to mRNA transcripts. While plant miRNAs match their targets with nearly-full complementarity leading to mRNA cleavage, miRNAs in most animals require only a short sequence called 'seed' to inhibit target translation. Recent findings showed that miRNAs in cnidarians, early-branching metazoans, act similarly to plant miRNAs, by exhibiting full complementarity and target cleavage; however, it remained unknown if seed-based regulation was possible in cnidarians. Here, we investigate the miRNA-target complementarity requirements for miRNA activity in the cnidarian *Nematostella vectensis*. We show that bilaterian-like complementarity of seed-only or seed and supplementary 3' matches are insufficient for miRNA-mediated knockdown. Furthermore, miRNA-target mismatches in the cleavage site decrease knockdown efficiency. Finally, miRNA silencing of a target with three seed binding sites in the 3' untranslated region that mimics typical miRNA targeting was repressed in zebrafish but not in *Nematostella* and another cnidarian, *Hydractinia symbiolongicarpus*. Altogether, these results unravel striking similarities between plant and cnidarian miRNAs supporting a possible common evolutionary origin of miRNAs in plants and animals.**

**Keywords** MicroRNA; Cnidaria; *Nematostella*
**Subject Categories** Evolution & Ecology; RNA Biology

## Introduction

miRNAs are endogenous post-transcriptional regulators that are abundant in diverse eukaryotic lineages (Bartel, 2004; Ameres and Zamore, 2013; Moran et al, 2017; Bartel, 2018). They have important roles in various biological processes and are essential for proper development of animals and plants (Bartel, 2018; Jones-Rhoades et al, 2006; Voinnet, 2009; Dexheimer and Cochella, 2020). miRNAs are

transcribed by RNA polymerase II into long primary transcripts that are processed into hairpin-structured precursor-miRNAs (pre-miR-NAs), which are later cleaved into short 20–22 nucleotides long duplexes. The duplexes are loaded into Argonaute (AGO) proteins that are a part of the RNA-Induced Silencing Complex (RISC), where only one strand is selected to remain loaded and the other is discarded (Kim et al, 2009). The loaded strand leads the RISC complex to a matching target and mediates its repression by inducing cleavage, translation inhibition or degradation by deadenylation of the mRNA (Jones-Rhoades et al, 2006; Jonas and Izaurralde, 2015).

The miRNA system is present in both plant and animal kingdoms, although a few major differences exist between them in the miRNA biogenesis pathway, mode of action and target recognition (Axtell et al, 2011; Moran et al, 2017). The biogenesis pathway in animals starts within the nucleus with the processing of the primary miRNA (pri-miRNA) by the microprocessor complex composed of RNase type III Drosha and its partner protein Pasha (known as DGCR8 in vertebrates) (Han et al, 2004a). The resulting pre-miRNA is transported by Exportin 5 into the cytoplasm where it gets cleaved into the mature miRNA by the RNase type III Dicer with the help of partner double-stranded RNA binding proteins such as Loqs and TRBP (Förstemann et al, 2005; Redfern et al, 2013; Wilson et al, 2015; Fareh et al, 2016; Jouravleva et al, 2022). In plants, both pri-miRNA and pre-miRNA are processed within the nucleus by DICER-LIKE1 (DCL1) assisted by its partner protein Hyponastic Leaves1 (HYL1) (Han et al, 2004b; Voinnet, 2009). Another difference between plant and animal miRNAs resides in their target recognition mode. In bilaterian animals, which include most known animal groups such as arthropods, nematodes, and vertebrates, miRNAs bind their targets with a short 5' sequence called the "seed" that includes only seven nucleotides, at positions 2–8 of the miRNA (Lai, 2002; Rhoades et al, 2002; Lewis et al, 2003; Brennecke et al, 2005; Ameres et al, 2007). Supplemental complementarity at the 3' end, mostly of positions 13–16, occurs in some cases, but it is not as frequent and considered less crucial for target recognition (Grimson et al, 2007; Bartel, 2009). The contribution of the supplemental complementarity to target binding seems to change considerably between cases (Becker et al, 2019; Bertolet et al, 2019). Target recognition restricted to seed or mediated via a seed match and supplemental complementarity with mismatches at positions 10 and 11 often

[1]Department of Ecology, Evolution and Behavior, Alexander Silberman Institute of Life Sciences, Faculty of Science, Hebrew University of Jerusalem, Jerusalem 9190401, Israel.
[2]Center for Chromosome Biology, School of Biological and Chemical Sciences, University of Galway, Galway, Ireland. [3]Department of Genetics, Alexander Silberman Institute of
Life Sciences, Faculty of Science, Hebrew University of Jerusalem, Jerusalem 9190401, Israel. ✉E-mail: yael.admoni@mail.huji.ac.il; yehu.moran@mail.huji.ac.il

leads to translational inhibition and deadenylation of the mRNA, mediated by the metazoan-specific GW182 protein family (called TNRC6 in vertebrates) (Hutvagner and Simard, 2008; Bartel, 2009; Iwakawa and Tomari, 2015). Contrastingly, plant miRNA-target recognition and activity require nearly-full complementarity that frequently results in AGO-mediated target cleavage between positions 10–11 of the miRNA, known as the cleavage site. Translational inhibition can also occur, but it still requires nearly-full complementarity for target recognition (Aukerman and Sakai, 2003; Chen, 2004; Gandikota et al, 2007; Brodersen et al, 2008; Li et al, 2013; Iwakawa and Tomari, 2013; Liu et al, 2014).

The above-mentioned differences led to the notion that the miRNA system evolved independently in plants and animals; however, recent studies have shown that the miRNA system in the model sea anemone *Nematostella vectensis*, as well as other cnidarian species, is more similar to plants than previously thought (Moran et al, 2014; Modepalli et al, 2018; Tripathi et al, 2022; Baumgarten et al, 2018; Li and Hui, 2023). Cnidaria, the sister group to Bilateria, diverged over 600 million years ago from the vast majority of animal clades, and is composed of sea anemones, corals, jellyfish, and hydroids (Erwin et al, 2011; Kayal et al, 2018). Cnidarians possess a miRNA system (Grimson et al, 2008), and share some highly conserved miRNAs, some of them are known to be crucial for cnidarian fitness and development (Modepalli et al, 2018; Praher et al, 2021; Fridrich et al, 2020, 2023). Interestingly, cnidarian miRNAs operate by binding their targets with nearly-full complementarity that leads to mRNA cleavage, in a similar manner to plant miRNAs (Moran et al, 2014). Furthermore, miRNAs in *Nematostella* and plants are methylated at the 3' end by the methyltransferase HUA ENHANCER1 (HEN1), which is essential to prevent miRNA degradation (Modepalli et al, 2018). Importantly, it was shown recently that *Nematostella* Hyl1-Like a, a homolog to plant-specific HYL1, also takes part in miRNA biogenesis, which suggests that it likely took part in miRNA biogenesis before the separation of plants and animals (Tripathi et al, 2022). Yet, despite these striking similarities to the miRNA pathway of plants, the cnidarian miRNA system also exhibits clear metazoan-specific features such as Drosha and Pasha homologs and a GW182 homolog (Moran et al, 2013; Mauri et al, 2017). The similarities to the bilaterian miRNA pathway raise the question whether cnidarian miRNAs might be able to target mRNAs via more restricted interactions focused on the seed region like their bilaterian counterparts.

To answer this question, we characterized the complementarity requirements between miRNAs and their targets in *Nematostella*. Using a reporter, we tested the efficiency of different complementarity patterns in promoting gene knockdown, such as the bilaterian-like seed match and a cleavage site mutated sequence. We utilized *TBP::mCherry* transgenic sea anemones that ubiquitously express mCherry fluorescent protein (Admoni et al, 2020) and injected into their zygotes different miRNA mimics to compare their effect on the expression of the fluorescent reporter.

# Results

## Bilaterian-like matches fail to repress gene expression in *Nematostella*

It was previously shown that injection of short hairpin RNAs (shRNAs) to *Nematostella* zygotes leads to efficient knockdown of chosen targets

(He et al, 2018). For our study, we designed a miRNA mimic (mimiR) based on an endogenous miRNA template to resemble native *Nematostella* miRNA precursors and better mimic their processing by Dicer and the strand selection by AGOs. We used the pre-miRNA sequence of Nve-miR-2022, a highly conserved miRNA among cnidarians (Moran et al, 2014; Nong et al, 2020; Praher et al, 2021), and changed the mature miRNA sequence to nearly fully match the mCherry transcript. The target site is located in the 3' UTR, since the majority of canonical miRNA sites are found in this region (Bartel, 2018). To be able to test the effect of different complementarity patterns on knockdown efficiency, we first generated mimiR with nearly-full complementarity to mCherry transcript (except for position 19, see "Methods"), that was later altered to resemble bilaterian miRNA binding sites that are based on seed match or mismatched in the cleavage site (Fig. 1A).

To test the impact of bilaterian-like miRNA-target complementarity pattern on transcript and protein levels, only the seed region at positions 2–8 of the miRNA was left matching the mCherry-encoding transcript while the rest of the sequence was changed. In addition, we generated a mimiR with supplemental matches to the seed at positions 13–16 (Fig. 1A). The mimiRs were injected into *TBP::mCherry* zygotes that were observed after 3 days, and mCherry transcript and protein levels were measured.

Interestingly, we observed that bilaterian-like mimiRs with "canonical site", i.e., matching their target only via the seed region, which is the most common class of miRNA in Bilateria (Bartel, 2018), cause no measurable knockdown of mCherry. The fluorescence of the embryos was similar to the negative control group injected with short hairpin RNA (shRNA) with no matches to *Nematostella* transcripts (Fig. 1B) and mCherry mRNA and protein levels showed no difference compared to the negative control; and were significantly higher than in the positive control groups (Fig. 1G,H; Table 1). Adding supplementary binding bases at the 3'-end of the miRNA, to resemble another type of common bilaterian targets, resulted in similar measurements, with both transcript and protein levels of mCherry remaining unaffected by the presence of the mimiR when compared to the control group (Fig. 1C,G,H; Table 1). These observations suggest that bilaterian-like matches have no gene knockdown effect in *Nematostella* similarly to plants, where they have very weak effect or none at all (Iwakawa and Tomari, 2013; Liu et al, 2014).

## Cleavage site mismatches interfere with miRNA activity

Next, we assessed the necessity of miRNA binding in the site of target cleavage. We mismatched positions 10–11 of the mimiR and compared mCherry levels to nearly-full complementarity control mimiR injection. Similarly to seed-restricted mimiRs, inhibition of cleavage site binding resulted in impaired miRNA activity, as mCherry fluorescence as well as transcript and protein levels showed no difference from the negative control (Fig. 1D,G,H; Table 1). These results are in accordance with plant miRNAs that also fail to induce target cleavage when central mismatches are introduced (Iwakawa and Tomari, 2013). Moreover, this experiment further validates the notion that *Nematostella* miRNAs promote target cleavage as the main mode of action (Moran et al, 2014).

After testing positions 10–11, we wished to test how a single mismatch in the cleavage site affects the knockdown efficiency. For this, we mismatched either position 10 or 11 separately and injected both variants to transgenic zygotes. Both mimiRs resulted in visibly lower mCherry

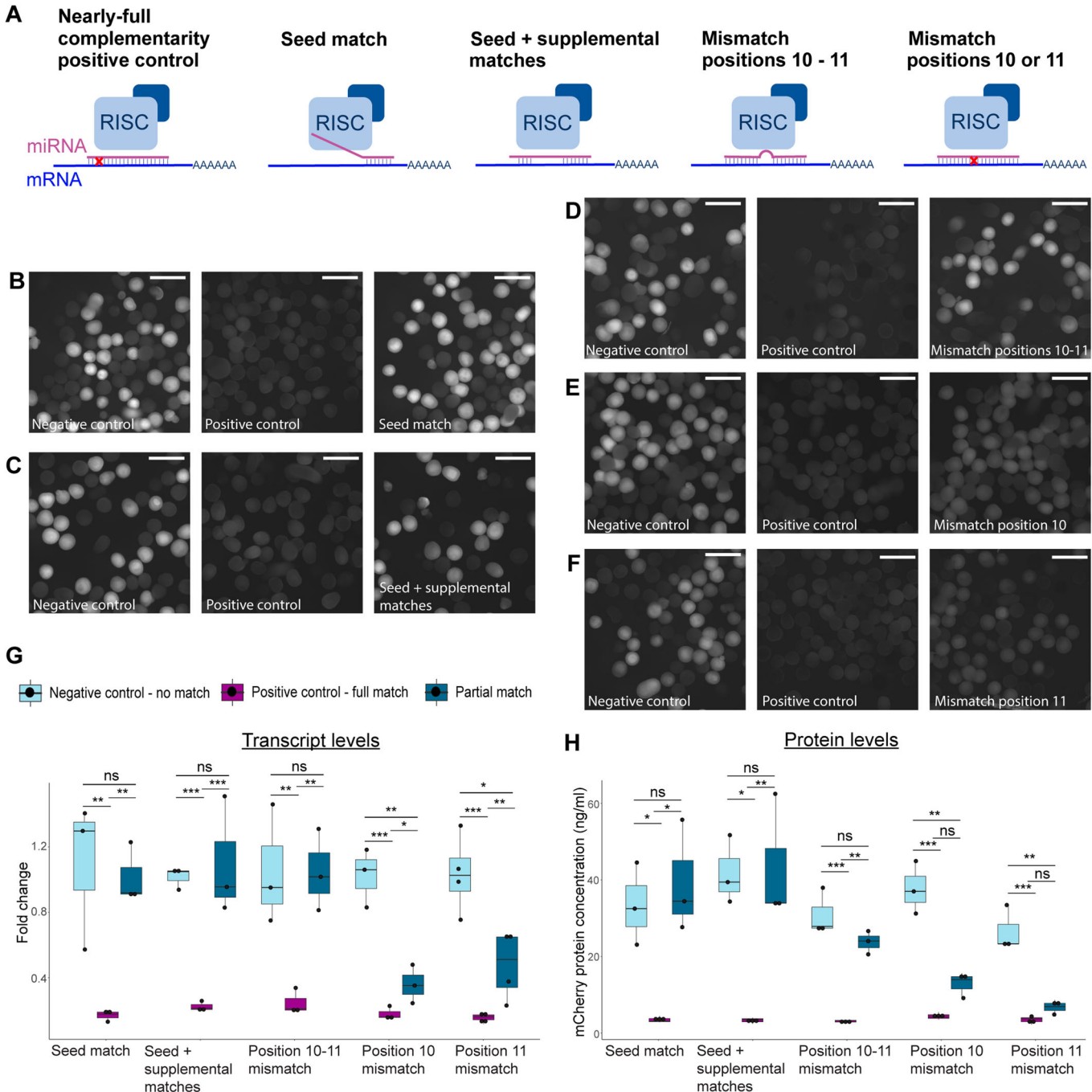

fluorescence. On the molecular level, mCherry transcript levels were significantly lower than the negative control, hence knockdown still occurred, but it was significantly weaker than the positive control inflicted repression (Fig. 1E–G; Table 1). On the protein level, despite a noticeable trend of higher protein levels compared to the positive control both in the ELISA measurement and the fluorescence of the transgenic animals, there was no statistically significant difference between protein concentration between a nearly-full match and a single position mismatch (Fig. 1E,F,H; Table 1). This raises the intriguing possibility that translational inhibition contributes to the silencing effect beyond the effect at the RNA level. Yet, such a claim requires further biochemical proof.

Recently, it was shown that a 9-amino acid loop structure in the PIWI domain of *Arabidopsis thaliana* AGO10 (atAGO10) is necessary for target recognition by the AGO-miRNA complex (Xiao et al, 2023a). This loop contributes to extensive complementarity in *Arabidopsis* and is disordered in human AGO2 (hsAGO2). The core structure of AGO-miRNA complex is highly conserved between plants and animals (Xiao et al, 2023b), which suggests that the small PIWI loop is a key characteristic in the slicing-based miRNA pathway. Thus, we looked at the predicted structure of the two *Nematostella* AGOs (NveAGO1 and NveAGO2) in search of a similar loop. Indeed, the predicted

**Figure 1. Silencing effects of different miRNA-target complementarity patterns in *Nematostella*.**

(A) Schematic representation of complementarity between designed mimiRs and mRNA target (B–F). mCherry fluorescence captured in *TBP::mCherry* heterozygote embryos, 3 days after injection with mimiRs. Negative control groups (left) injected with shRNA with no match in *Nematostella* genome, displaying noticeable fluorescence. Positive control (middle) groups injected with nearly-full complementarity mimiR displaying no visible mCherry fluorescence. Groups injected with partial match mimiRs (right) displaying varying levels of fluorescence with (B) seed match (C) seed + supplemental matches and (D) mismatched positions 10–11 showing fluorescence similar to negative control and (E) mismatched position 10 and (F) mismatched position 11 showing intermediate fluorescence. Scale bars represent 500 μm. (G). mCherry transcript fold change 3 days after injection with different mimiRs, measured by RT-qPCR. Significance is shown for pairwise comparisons (one-way ANOVA with Tukey's HSD post hoc test, $n = 3$ biological replicates for all groups but mismatched position 11 that has $n = 4$ biological replicates). $P$ values for each group are mentioned in this order: positive control–partial match, negative control–partial match, negative control–positive control. Seed match: 0.001295, 0.999682, 0.001271. Seed + supplemental matches: 0.000176, 0.958615, 0.000208. Position 10–11 mismatch: 0.002196, 0.998366, 0.002302. Position 10 mismatch: 0.040092, 0.005294, 0.000392. Position 11 mismatch: 0.004229, 0.018366, 0.0000674. (H) mCherry protein concentration 3 days after injection with different mimiRs, measured by ELISA assay. Significance is shown for pairwise comparisons (one-way ANOVA with Tukey's HSD post hoc test, $n = 3$ biological replicates). $P$ values for each group are mentioned in this order: positive control–partial match, negative control–partial match, negative control–positive control. Seed match: 0.013808, 0.779224, 0.030149. Seed + supplemental matches: 0.009418, 0.982099, 0.0114. Position 10–11 mismatch: 0.001664, 0.146316, 0.00033. Position 10 mismatch: 0.125255, 0.001186, 0.000234. Position 11 mismatch: 0.525893, 0.001273, 0.000544. Data information: in (G, H) box plots show median, the lower and upper bounds correspond to the 25th and 75th percentiles and whiskers extend to maximum and minimum values. The statistically significant difference is represented by: *$P$ value < 0.05, **$P$ value < 0.01, ***$P$ value < 0.001, ns not significant. Source data are available online for this figure.

structures indicate the presence of a loop in *Nematostella* AGOs that aligns with the atAGO10 loop, although when using the prediction method for hsAGO2 structure, the loop appears as well (Fig. EV1A,B). Hence, structural work is necessary to resolve the exact structure of *Nematostella* AGOs to determine if the PIWI loop is conserved between plants and cnidarians.

## Multiple seed match sites in the 3′ UTR are inefficient for miRNA silencing activity in Cnidaria

Single miRNA binding sites in bilaterians are capable of modulating protein target levels in a significant manner as demonstrated experimentally in *Drosophila* and zebrafish (Brennecke et al, 2005; Choi et al, 2007). Nevertheless, many bilaterian miRNAs exhibit more than one site for each target they regulate (Grimson et al, 2007). Since multiple binding sites on the same target transcript can provide together synergic rather than additive repression effects (Briskin et al, 2020), we designed a mCherry-encoding mRNA that harbors three seed match sites in its 3′ UTR (Fig. 2A). The mRNA was injected into wild-type (WT) *Nematostella* zygotes along with the previously used seed match mimiR. A new nearly fully matching positive control mimiR was designed since the original sequence could potentially partially bind the three seed sites. After 24 h from injection, mCherry fluorescence was weaker in the positive control group compared to the seed match group (Fig. 2B). In accordance, protein concentrations were similar between seed match mimiR and negative control, both significantly higher than the positive control (Fig. 2C; Table 1). This result shows that increasing the number of target sites in the 3′ UTR does not improve the efficiency of seed match miRNAs and further validates that bilaterian-like matches are ineffective in *Nematostella*.

Next, the mRNA silencing by mimiRs was tested in *Hydractinia symbiolongicarpus*, a colonial cnidarian and member of Medusozoa, which separated from Anthozoa, the group that includes *Nematostella*, about 560 million years ago (Khalturin et al, 2019). shRNA silencing tool was shown to be effective in *Hydractinia* (DuBuc et al, 2020), making it possible to test this experimental design in another cnidarian. In a similar design to the experiment in *Nematostella*, mCherry mRNA with the same 3′ UTR with three seed match sites was injected into *Hydractinia* zygotes along with mimiRs or negative control. mCherry protein concentrations were

compared 24 h after injection, revealing that the nearly-full complementarity mimiR was effective in knocking down mCherry expression in *Hydractinia* embryos (Fig. EV2; Table 1). In comparison, the effect of the seed match mimiR was inconclusive, due to high variability of mCherry expression in the negative control and seed match groups compared to the positive control. Following these results, the experimental method was changed to measuring normalized fluorescence of GFP, which has better expression level and visibility in *Hydractinia*, and was also encoded by mRNA with the same three seed match sites and identical 3′ UTR. The injected animals were observed under the fluorescent microscope and GFP levels were measured and normalized to a fluorescent red tracer dye that was included in the injection mix. Similarly to the previous observations in *Nematostella* and *Hydractinia*, GFP fluorescence levels following injection of nearly-full complementarity mimiR were significantly lower than both the seed match and the negative control groups after 24 h (Fig. 3A,B; Table 1). In contrast, the difference between the negative control and the seed groups was not significant. In addition, equivalence test between these groups with bounds of −0.19 and 0.19 is significant ($P$ value = 0.047) and the null significance hypothesis that the effect is equal to zero is not significant, supporting that the effect caused by negative control and the seed match mimiR is similar. These results show that in *Hydractinia* as well as *Nematostella*, the miRNA mechanism is based on nearly-full complementarity between miRNAs and their targets, while seed match alone is insufficient in mediating target knockdown.

## Validation of miRNA-mimics loading and activity

To confirm that the mimiRs are being loaded onto *Nematostella* AGOs, immunoprecipitation (IP) was performed using specific antibodies for each AGO (Fridrich et al, 2020), following injection of mimiRs into WT zygotes. Small RNAs (sRNAs) were sequenced from immunoprecipitated samples of NveAGO1, NveAGO2, and rabbit IgG as control and reads mapping to mimiR sequences were quantified. The results show that all three tested mimiRs, full match, seed match and seed with supplementary matches, were effectively loaded onto *Nematostella* AGOs (Fig. 4A–C). Notably, the loading was more efficient onto NveAGO2. This might be because NveAGO1 typically loads more specific miRNAs than NveAGO2 (Fridrich et al, 2020). Moreover, the

**Table 1.   Pairwise comparisons.**

| | | Difference of means | 95% CI | Adjusted *P* value |
|---|---|---|---|---|
| Transcript | Full match–seed match | 2.68132485 | 1.45547, 3.907179 | 0.001295 |
| | No match–seed match | −0.0096041 | −1.23546, 1.21625 | 0.999682 |
| | No match–full match | −2.690929 | −3.91678, −1.46508 | 0.001271 |
| | Full match–seed + supplementary match | 2.32389005 | 1.583794, 3.063986 | 0.000176 |
| | No match–seed + supplementary match | 0.06702847 | −0.67307, 0.807124 | 0.958615 |
| | No match–full match | −2.2568616 | −2.99696, 1.51677 | 0.000208 |
| | Full match–position 10-11 mismatch | 2.16048876 | 1.068762, 3.252215 | 0.002196 |
| | No match–position 10-11 mismatch | 0.01938205 | −1.07234, 1.111108 | 0.998366 |
| | No match– full match | −2.1411067 | −3.23283, 1.04938 | 0.002302 |
| | Full match–position 10 mismatch | 1.011219 | 0.056283, 1.966155 | 0.040092 |
| | No match–position 10 mismatch | −1.589497 | −2.54443, −0.63456 | 0.005294 |
| | No match–full match | −2.600716 | −3.55565, −1.64578 | 0.000392 |
| | Full match–position 11 mismatch | 1.595208 | 0.589205, 2.601211 | 0.004229 |
| | No match–position 11 mismatch | −1.238501 | −2.2445, −0.2325 | 0.018366 |
| | No match–full match | −2.833709 | −3.83971, −1.82771 | 0.0000674 |
| Protein | Full match–seed match | −35,790.49 | −62,095.774, −9485.21 | 0.013808 |
| | No match–seed match | −5892.439 | −32,197.724, 20,412.85 | 0.779224 |
| | No match–full match | 29,898.05 | 3592.766, 56,203.34 | 0.030149 |
| | Full match–seed + supplementary match | −40,122.849 | −67,276.71, −12,969 | 0.009418 |
| | No match–seed + supplementary match | −1604.372 | −28,758.24, 25,549.49 | 0.982099 |
| | No match–full match | 38,518.477 | 11,364.61, 65,672.34 | 0.0114 |
| | Full match–position 10-11 mismatch | −20,684.861 | −30,598.14, −10,771.6 | 0.001664 |
| | No match–position 10-11 mismatch | 7165.728 | −2747.55, 17,079 | 0.146316 |
| | No match–full match | 27,850.588 | 17,937.31, 37,763.87 | 0.00033 |
| | Full match–position 10 mismatch | −8511.601 | −19,674.01, 2650.81 | 0.125255 |
| | No match–position 10 mismatch | 24,822.125 | 13,659.71, 35,984.54 | 0.001186 |
| | No match–full match | 33,333.726 | 22,171.32, 44,496.14 | 0.000234 |
| | Full match–position 11 mismatch | −3368.935 | −12,422.04, 5684.172 | 0.525893 |
| | No match–position 11 mismatch | 19,866.46 | 10,813.35, 28,919.57 | 0.001273 |
| | No match–full match | 23,235.395 | 14,182.29, 32,288.5 | 0.000544 |
| mRNA injections *Nematostella* - protein | Full match–seed match | 25,544.143 | 15,900.65, 35,187.65 | 0.0029098 |
| | No match–seed match | 1314.02 | −7896.52, 10,524.55 | 0.8971442 |
| | No match–full match | −24,230.127 | −31337.1, −17,123.2 | 0.0012039 |
| mRNA injections *Hydractinia* - protein | Full match–seed match | −3192.706 | −9279.660, 2894.249 | 0.3124345 |
| | No match–seed match | 4869.152 | −1217.802, 10,956.11 | 0.1081857 |
| | No match–full match | 8061.858 | 1974.904, 14,148.81 | 0.0155945 |
| mRNA injections *Hydractinia* - GFP fluorescence | Full match–seed match | 0.174 | 0.089, 0.259 | 0.0000269 |
| | No match–seed match | −0.099 | −0.228, 0.031 | 0.167 |
| | No match–full match | −0.273 | −0.395, −0.151 | 0.000008 |
| mRNA injections zebrafish - protein | Full match–seed match | 4341.821 | −5918.182, 14,601.8253 | 0.4923792 |
| | No match–seed match | 13,842.273 | 3582.2694, 24,102.277 | 0.0111299 |
| | No match–full match | 9500.452 | −759.5521, 19,760.455 | 0.0689794 |
| mRNA injections zebrafish - mCherry fluorescence | Full match–seed match | 0.3259888 | −0.1327552, 0.7847328 | 0.1583761 |
| | No match–seed match | 0.6482941 | 0.1895502, 1.1070382 | 0.0116010 |
| | No match–full match | 0.3223053 | −0.1364386, 0.7810494 | 0.1534842 |

Results of pairwise comparisons between injected groups. Including difference of means for mCherry transcript (ΔCt), protein concentration or normalized fluorescence, confidence interval and adjusted *P* value for multiple comparisons. *P* values were adjusted with Tukey's HSD or Games–Howell post hoc test. Full match refers to positive control group injected with nearly-full complementarity mimiR. No match refers to negative control group injected with shRNA with no match in *Nematostella* genome.

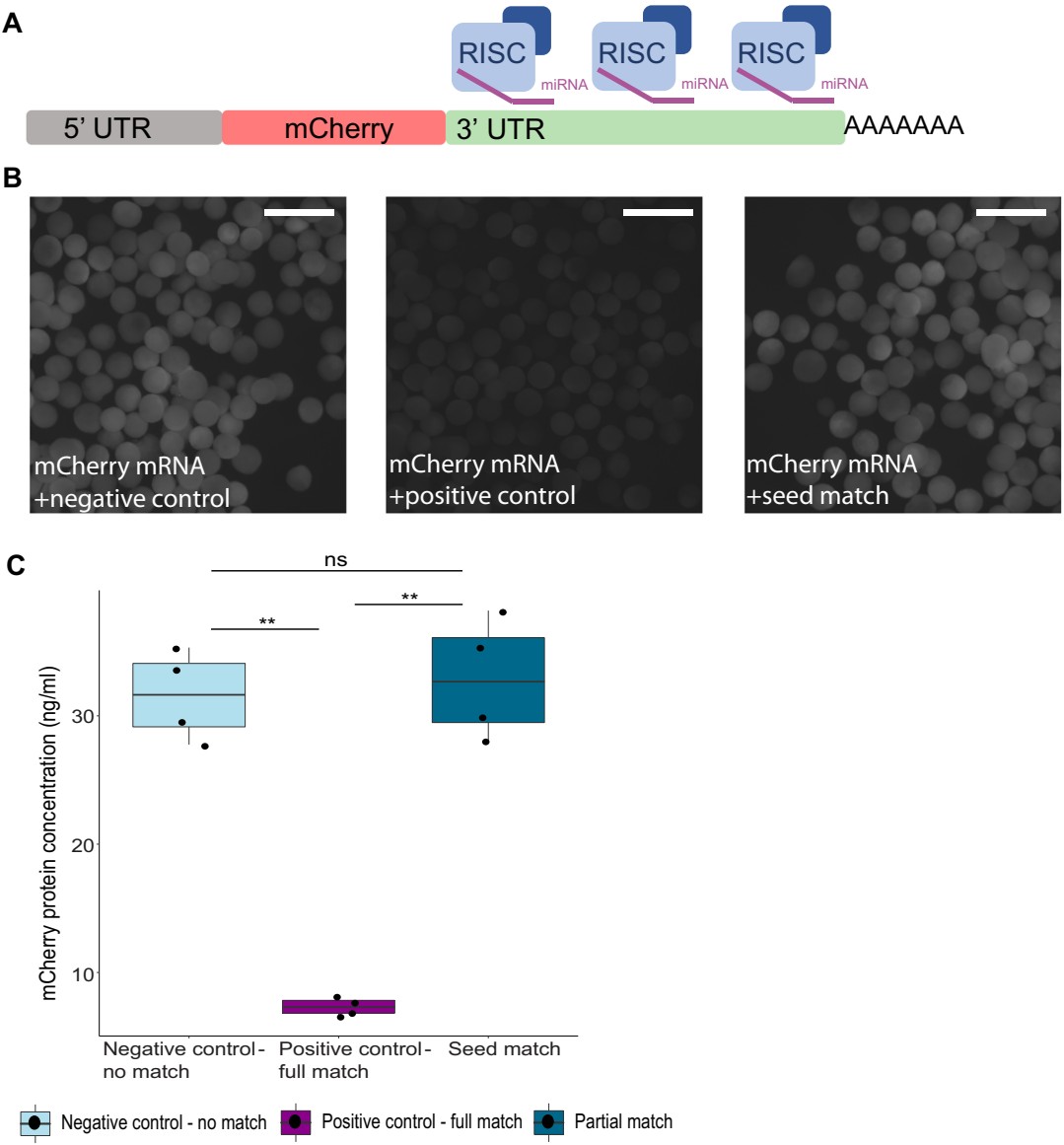

**Figure 2. mRNA with multiple seed match sites is not silenced by seed match mimiR in *Nematostella*.**

(A) Schematic representation of injected in vitro transcribed mCherry mRNA, containing EF1α kozak sequence followed by mCherry-encoding transcript containing three seed match sites in its 3′ UTR. (B) mCherry fluorescence observed in WT *Nematostella* embryos, 24 h after injection with mCherry mRNA combined with a different mimiR in each panel. Negative control group (left) injected with shRNA with no match in *Nematostella* genome, displaying noticeable fluorescence. Positive control (middle) group injected with nearly-full complementarity mimiR displaying no visible mCherry fluorescence. Seed match group (right) injected with mimiR matching the three seed sites in the 3′ UTR showing fluorescence similar to negative control. Scale bars represent 500 μm. (C) mCherry protein concentration 24 h after injection with mCherry mRNA combined with different mimiRs to *Nematostella* zygotes. Measured by ELISA assay. Significance is shown for pairwise comparisons (one-way Welch's ANOVA with Games–Howell post hoc test, $n = 4$ biological replicates). P values are: positive control–partial match 0.0029098, negative control–partial match 0.8971442, negative control–positive control 0.0012039. Data information: in (C) box plots show median, the lower and upper bounds correspond to the 25th and 75th percentiles and whiskers extend to maximum and minimum values. Statistically significant difference is represented by: **$P$ value < 0.01, ns not significant. Source data are available online for this figure.

full match mimiR was loaded less efficiently than the others, despite causing the strongest knockdown of the injected mRNA transcript, strengthening the conclusion that bilaterian-like matches are ineffective in *Nematostella*. The reason for the lower number of full match mimiR reads could be inaccurate processing by Dicer, as some of the guide reads included parts of the mimiR precursor loop (Fig. EV3A). In contrast, mimiRs seed match and seed with supplementary matches

show more homogenous processing (Fig. EV3B,C). As expected, reads corresponding to the guide strand were more abundant than the star strand for all three mimiRs, which suggests that basing the mimiR design on miR-2022 template was mostly successful.

To validate the ability of the sRNAs to bind to the sites and promote silencing in a bilaterian system, the target mRNA was injected with shRNA/mimiR into zebrafish embryos. The injection

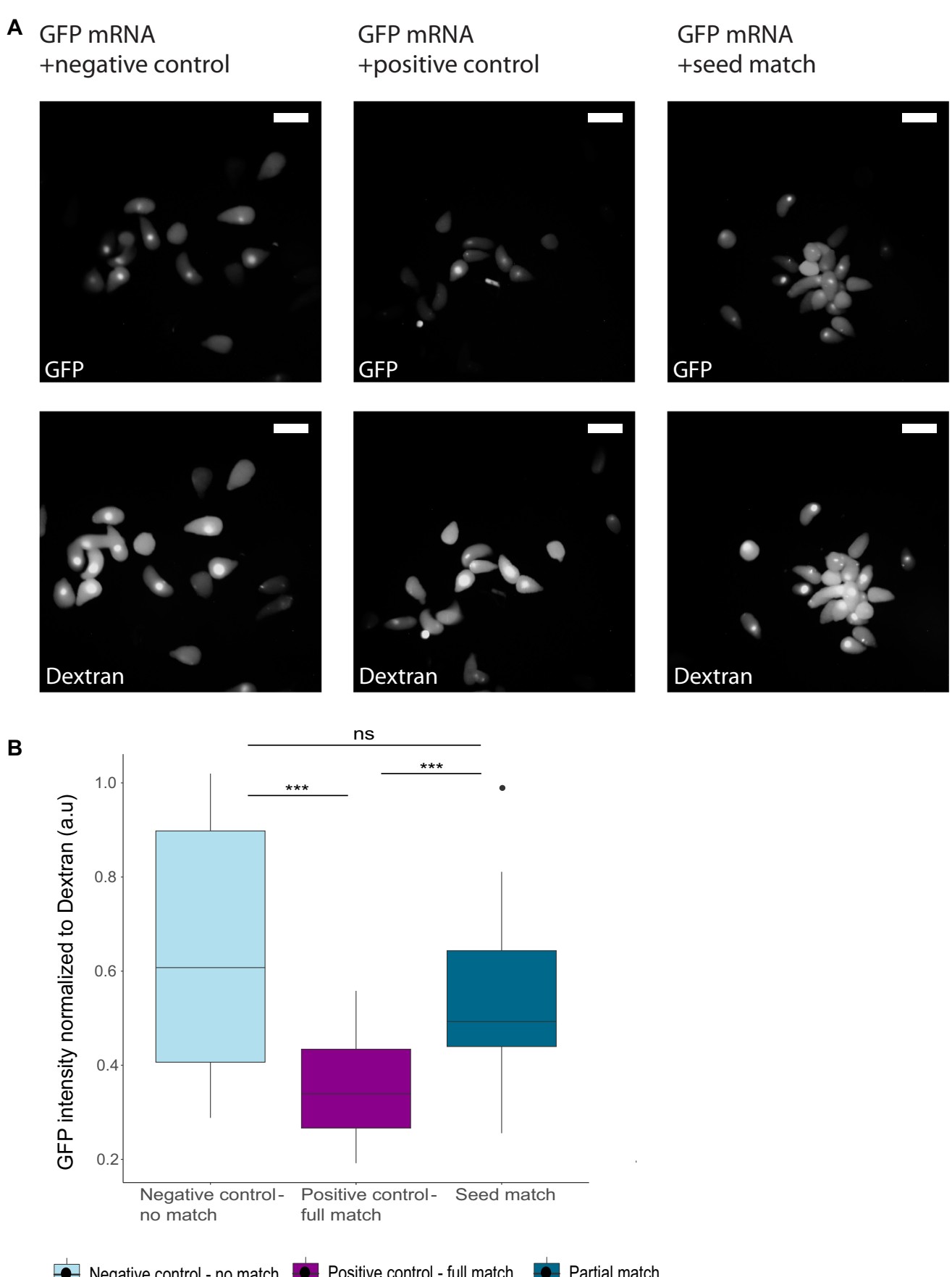

**Figure 3. mRNA with multiple seed match sites is not silenced by seed match mimiR in *Hydractinia*.**

(A) Fluorescence observed in *Hydractinia* embryos, 24 h after injection with GFP mRNA combined with different mimiRs and red dextran tracer as fluorescence intensity control. Top pictures show GFP fluorescence in the different treatments: Negative control group (left) injected with shRNA with no match to GFP mRNA, displaying noticeable fluorescence. Positive control (middle) group injected with nearly-full complementarity mimiR displaying weaker GFP fluorescence. Seed match group (right) injected with mimiR matching the three seed sites in the 3′ UTR showing stronger fluorescence than the positive control group. Bottom pictures show dextran tracer fluorescence that was used for normalization of GFP expression. Scale bars represent 500 μm. (B) GFP fluorescence normalized to red dextran tracer 24 h after injection with GFP mRNA combined with different mimiRs. Significance is shown for pairwise comparisons (one-way Welch's ANOVA with Games–Howell post hoc test, Negative control = 30 embryos, Positive control = 24 embryos, Seed match = 31 embryos, $n = 3$ biological replicates). P values are: positive control–partial match 0.0000269, negative control–partial match 0.167, negative control–positive control 0.000008. Data information: in (B) box plots show median, the lower and upper bounds correspond to the 25th and 75th percentiles and whiskers extend to maximum and minimum values. Statistically significant difference is represented by: ***P value < 0.001, ns not significant. Source data are available online for this figure.

was in the same manner as to *Nematostella*, with the addition of an mRNA encoding sfGFP without miRNA binding sites, to account for the variability of expression efficiency in the zebrafish embryos. Following 10 h from injection, the embryos showed a difference in mCherry fluorescence between the groups, with the seed match group exhibiting the weakest fluorescence (Fig. 5A). Both in the protein and the fluorescence level, a significant difference was found between the negative control and the seed match groups, hence validating the efficiency of the seed match mimiR in binding and repressing its target in a bilaterian animal (Fig. 5B,C; Table 1). Moreover, no significant difference was found between the full match mimiR group and the other treatments, indicating that the target was not efficiently cleaved despite the extensive complementarity. This result is in accordance with the fact that zebrafish lack efficient slicing activity by AGO2 (drAGO2) due to two specific point substitutions specific to teleost fishes (Chen et al, 2017). In conclusion, the results of this experiment validate the experimental approach of co-injection of seed match or nearly-full complementarity mimiR and the mRNA target since the results coincide with the literature about zebrafish miRNA pathway.

## Discussion

In this study, we show through in-vivo assays that cnidarian miRNAs act similarly to those of plants in terms of the complementarity requirements to their targets to induce efficient gene repression. We show that bilaterian matches that rely either solely on seed matches or seed matches with supplementary 3′ matches fail to perform measurable gene repression in *Nematostella*. Importantly, seed match also failed to cause gene knockdown in *Nematostella* and *Hydractinia* when the mRNA target included three binding sites. This implies that the previously described nearly-perfect plant-like matches between cnidarian miRNAs and their targets (Moran et al, 2014) are the major mode of interaction. In addition, bilaterian-like matches are probably not functional in Cnidaria, as they are not functional in plants (Iwakawa and Tomari, 2013; Liu et al, 2014). Our experimental validation supports the evolutionary scenario that miRNA targeting based on seed match is a bilaterian innovation, suggested to contribute to the expansion of regulatory networks by allowing a single miRNA to bind hundreds of targets (Moran et al, 2017). It is noteworthy that in plants despite having full complementarity to the targets, the seed region is still crucial for target recognition and mismatches in positions 1–8 lead to a decrease in silencing efficiency (Mallory et al, 2004; Bartel, 2004; Liu et al, 2014), which could potentially also be the case for *Nematostella* and *Hydractinia* miRNAs.

Target cleavage is known to be the main mechanism for miRNA activity in plants, and we show that *Nematostella* miRNAs are dysfunctional without a cleavage site match, suggesting that cleavage is the ancestral miRNA mode of action. This has been discussed in relation to the ancient RNA interference (RNAi) system for defense against invasive nucleic acids, such as transposons and viruses, that operates by binding of short interfering RNAs (siRNAs) to foreign RNA targets and eliminating their expression by cleaving them. It is a probable evolutionary scenario that the miRNA system evolved from the RNAi defense system (Cerutti and Casas-Mollano, 2006). Reduced knockdown efficiency due to mismatch of one nucleotide at the cleavage site could be due to different conformation of AGO-miRNA that changes the cleavage efficiency (Sheu-Gruttadauria and MacRae, 2017). Some *Nematostella* miRNAs naturally exhibit a mismatch to their target in position 10 or 11: position 10 was found mismatched in 67 miRNA-target pairs, while position 11 was mismatched in 41 pairs among degradome-verified targets (Moran et al, 2014). It is possible that the natural mismatches are selected for weaker knockdown of their targets, as weaker repression might be beneficial for some regulatory roles.

In addition to target cleavage, translational inhibition, which is the common miRNA mechanism in bilaterians, was also exhibited in plants (Aukerman and Sakai, 2003; Chen, 2004; Gandikota et al, 2007; Brodersen et al, 2008; Li et al, 2013). It was shown in *Arabidopsis thaliana* that central mismatches abolish target cleavage, although still allow translational inhibition when the target site is in the 5′ UTR (Iwakawa and Tomari, 2013). The extent to which translational inhibition is occurring in *Nematostella* is still unknown, although it was shown that *Nematostella* GW182/TNRC6 homolog could promote mRNA decay and translational repression when expressed in human cells (Mauri et al, 2017). Based on our experiments, it seems that translational inhibition or mRNA decay did not occur when central mismatches prevented target cleavage. Yet, we cannot exclude the scenario that this mechanism is active in *Nematostella* due to the conservation of GW182 but requires a different complementarity pattern or a different number of sites.

The origin of the miRNA system has been under debate for many years, the two competing scenarios being that miRNAs are ancestral and existed before the separation of plants and animals, or that they evolved independently in multiple lineages (Axtell et al, 2011; Moran et al, 2017). Notably, many eukaryotic lineages lack a miRNA system, which could indicate that the miRNA system evolved convergently multiple times (Tarver et al, 2012; Moran et al, 2017; Burki et al, 2020). Alternatively, the miRNA system could have been lost multiple times. This notion is supported by evidence showing that RNAi-based systems could be lost, as it was

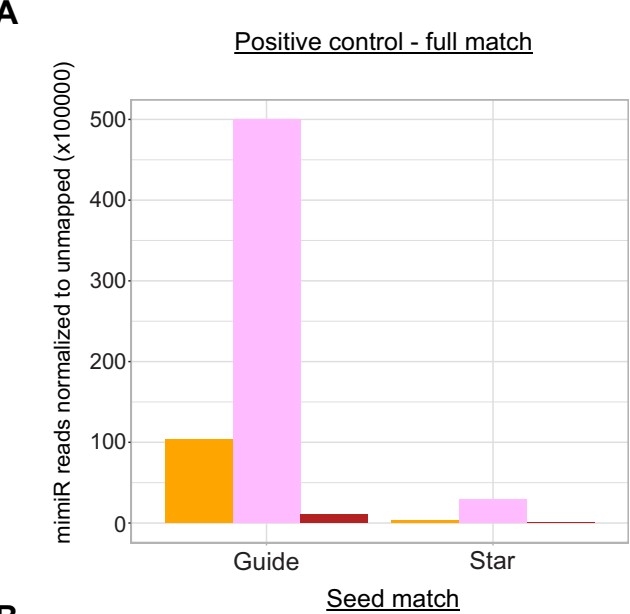

**A** Positive control - full match

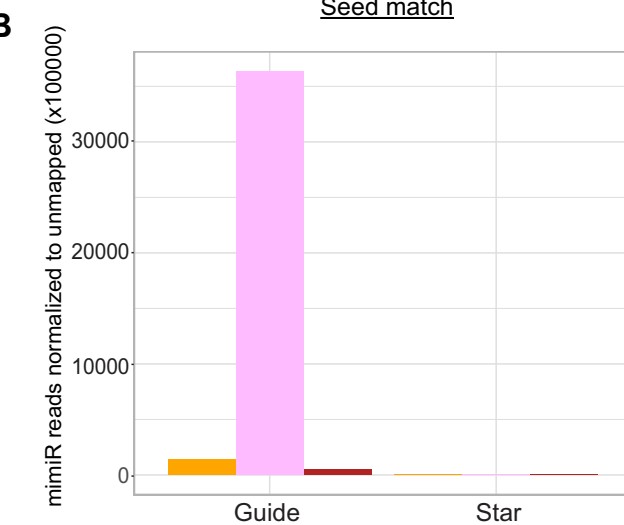

**B** Seed match

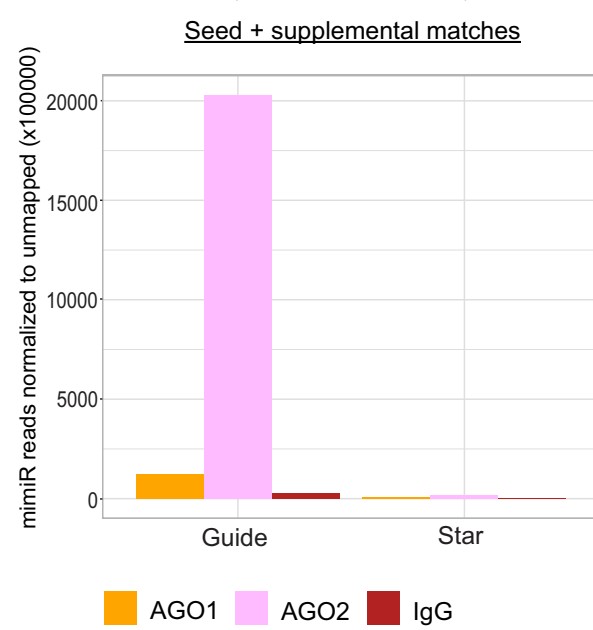

**C** Seed + supplemental matches

AGO1    AGO2    IgG

Figure 4.   mimiRs are loaded onto *Nematostella* AGOs.

(**A**) Reads mapping to nearly-full match mimiR (positive control) following NveAGO IP ($n = 2$ biological replicates). (**B**) Reads mapping to seed match mimiR following NveAGO IP ($n = 2$ biological replicates). (**C**) Reads mapping to seed + supplemental matches mimiR following NveAGO IP ($n = 2$ biological replicates).

lost in fungi, and miRNAs were lost in placozoans and probably in ctenophores (Grimson et al, 2008; Drinnenberg et al, 2011; Maxwell et al, 2012). It is also possible that due to limited analyses in other eukaryotic lines, the miRNA pathway is more prevalent in eukaryotes than currently known. More recent studies reveal miRNAs and miRNA-machinery in Amoebozoa and other single cell eukaryotes (Bråte et al, 2018; Edelbroek et al, 2024) and miRNA system components were also recently discovered in Asgard archaea (Bastiaanssen et al, 2024), demonstrating the limited knowledge we currently hold on this topic.

The lack of sequence homology between miRNAs of plants and animals is often used in support of the convergence hypothesis (Tarver et al, 2012). It could be claimed that if plant and animal miRNAs have an ancestral origin, then some ancestral miRNAs that would have existed in their last common ancestor would be shared between them. However, it should be noted that despite a few documented cases (Grimson et al, 2008; Lin et al, 2016; Tsuzuki et al, 2016; Praher et al, 2021), miRNA sequence conservation is uncommon within lineages (Robinson et al, 2013; Voshall et al, 2017; Moran et al, 2017).

Overall, the results of this study reveal important similarities between plants and cnidarians in the complementarity require-ments between miRNAs and their targets that join previous findings such as the conservation of HEN1 and HYL1 between plants and cnidarians, and together could provide support for a common origin of miRNA regulation before the separation of plants and animals (Fig. 6).

# Methods

**Reagents and tools table**

| Reagent/resource | Reference or source | Identifier or catalog number |
|---|---|---|
| **Experimental models** | | |
| Wild-type Rhode River, Maryland lab strain (*N.vectensis*) | Moran Lab | |
| TBP::mCherry (*N.vectensis*) | Admoni et al, 2020 | |
| Wild-type Male 291-10 and Female 296-10 (*H. symbiolongicarpus*) | Frank lab | |
| Wild-type AB/TL (*D. rerio*) | Rabani lab | |
| **Antibodies** | | |
| rabbit IgG antibody | Merck Millipore | I5006-10MG |
| NveAGO1 antibody | GenScript, Fridrich et al, 2020 | |
| NveAGO2 antibody | GenScript, Fridrich et al, 2020 | |

| Reagent/resource | Reference or source | Identifier or catalog number |
|---|---|---|
| **Oligonucleotides and other sequence-based reagents** | | |
| PCR primers | Integrated DNA Technologies, this study | Table 2 |
| shRNA/mimiR templates | Integrated DNA Technologies, this study | Table 2 |
| mRNA template | This study | Table 2 |
| **Chemicals, enzymes, and other reagents** | | |
| L-Cysteine | Merck Millipore | 1028380100 |
| dextran Alexa Fluor 4.88 | Thermo Fisher Scientific | D22910 |
| Red sea salt | Red sea | |
| phenol red | New England Biolabs | M0535L |
| Pronase | Merck Millipore | P5147 |
| Tri-Reagent® | Merck Millipore | T9424 |
| Turbo DNase | Thermo Fisher Scientific | AM1907 |
| RNase-free water | Merck Millipore | W4502-1L |
| Qubit™ RNA BR Assay Kit | Thermo Fisher Scientific | Q10210 |
| Deionized formamide | Merck Millipore | F9037-100ML |
| Glycogen | Merck Millipore | 10901393001 |
| AmpliScribe™ T7-Flash™ Transcription kit protocol | Lucigen | ASF3507 |
| Quick-RNA MiniPrep Kit | Zymo Research | R1054 |
| iScript™ cDNA Synthesis Kit | Bio-Rad | 1708891 |
| Fast SYBR® Green Master Mix | Thermo Fisher Scientific | AB-4385612 |
| Halt™ Protease Inhibitor cocktail | Thermo Fisher Scientific | 87786 |
| Pierce™ BCA Protein Assay Kit | Thermo Fisher Scientific | 23227 |
| RFP ELISA kit | Cell Biolabs | AKR-122 |
| SMARTer® RACE 5'/3' Kit | Takara Bio | 634858 |
| Advantage® 2 Polymerase | Takara Bio | 639201 |
| pGEM®-T Easy plasmid | Promega | A1360 |
| HighYield T7 Cap 1 AG (3'-OMe) mRNA Synthesis Kit (m5CTP) | Jena Bioscience | RNT-122-S |
| HiScribe T7 mRNA Kit with CleanCap Reagent AG | New England Biolabs | E2080S |
| RNA clean & concentrator 25 | Zymo Research | R1017 |
| *Escherichia coli* Poly(A) Polymerase | New England Biolabs | M0276S |
| RNA clean & concentrator 5 | Zymo Research | R1013 |
| SapI restriction enzyme | New England Biolabs | R0569S |
| HiScribe SP6 RNA synthesis kit | New England Biolabs | E2070S |
| Protein A Magnetic Beads | MedChem Express | HY-K0203 |
| Protease inhibitor cOmplete ULTRA tablets | Merck Millipore | 5892970001 |

| Reagent/resource | Reference or source | Identifier or catalog number |
|---|---|---|
| Protease Inhibitor Cocktail Set III, EDTA-Free | Merck Millipore | 539134-1 ML |
| Murine RNAse inhibitor | New England Biolabs | M0314L |
| NP-40 | Sigma-Aldrich | NP40S-100ML |
| 15% Criterion TBE-Urea Gel 12 + 2 45 μl | Bio-Rad | 3450091 |
| NEBNext Multiplex Small RNA Library Prep Set for Illumina kit | New England Biolabs | NEB-E7300S |
| T4 RNA Ligase 2 truncated KQ | New England Biolabs | M0373S |
| Certified™ Low Range Ultra Agarose | Bio-Rad | 1613107 |
| orange gel loading dye | New England Biolabs | B7022S |
| NucleoSpin Gel and PCR Clean-up | Macherey-Nagel | MAN-740609.50 |
| Qubit™ dsDNA HS (high sensitivity) Assay Kit | Thermo Fisher Scientific | Q32851 |
| High Sensitivity D1000 ScreenTape | Agilent Technologies | 5067-5585 |
| NextSeq 1000/2000 P2 Reagents (100 Cycles) v3 | Illumina | 20046811 |
| **Software** | | |
| CurveExpert Basic 2.2.3 | Hyams Development | |
| miRDeep2 | Friedländer et al, 2012 | |
| Bowtie1 (version 1.3.1 | Langmead et al, 2009 | |
| NIS-Elements Imaging Software | Nikon | |
| ImageJ software | Schindelin et al, 2012 | |
| ColabFold | Mirdita et al, 2022; Jumper et al, 2021 | |
| RCSB PDB Pairwise Structure Alignment tool | RCSB.org, Berman et al, 2000 | |
| PyMOL | Schrödinger, LLC | |
| Rstudio 2021.09.0 | | |
| Primer3 version 0.4.0 | Untergasser et al, 2012 | |
| **Other** | | |
| Eclipse Ti-S Inverted Research Microscopes | Nikon | |
| Intensilight fiber fluorescent illumination system | Nikon | |
| NT88-V3 Micromanipulator Systems | Narishige | |
| Epoch Microplate Spectrophotometer | BioTek Instruments | |
| StepOnePlus Real-Time PCR System v2.2 | ABI, Thermo Fisher Scientific | |
| Pestle mixer | Argos Technologies | A0001 |
| NextSeq2000 | Illumina | |
| SMZ18 stereomicroscope | Nikon | |
| DS-Qi2 SLR camera | Nikon | |

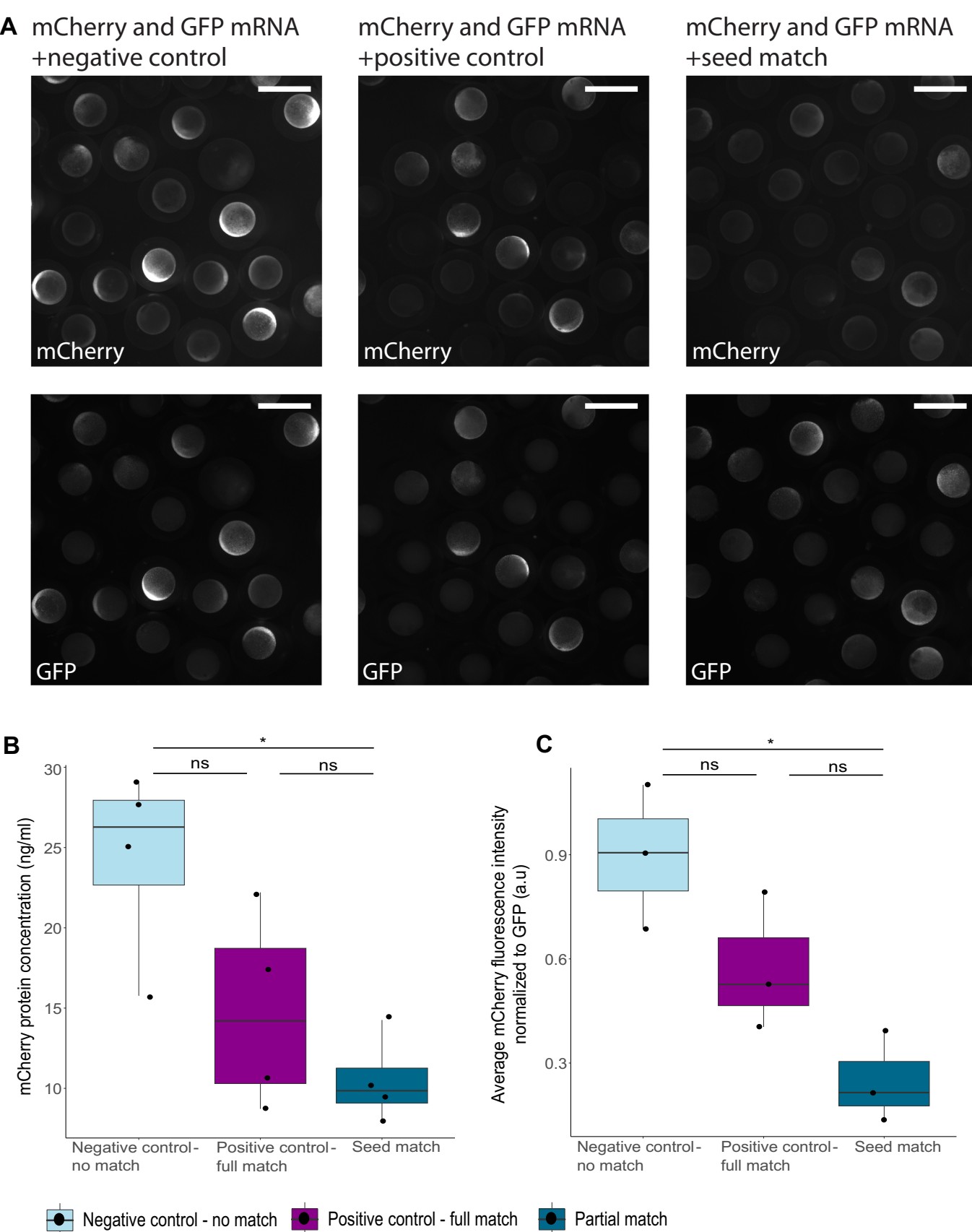

**Figure 5.  mRNA with multiple seed match sites is repressed by seed match mimiR in zebrafish.**

(A) Fluorescence observed in zebrafish embryos, 10 h after injection with mCherry mRNA combined with different mimiRs and sfGFP mRNA as fluorescence intensity control. Top pictures show mCherry fluorescence in the different treatments: Negative control group (left) injected with shRNA with no match to mCherry mRNA, displaying noticeable fluorescence. Positive control (middle) group injected with nearly-full complementarity mimiR displaying weaker mCherry fluorescence. Seed match group (right) injected with mimiR matching the three seed sites in the 3′ UTR showing the weakest fluorescence of the groups. Bottom pictures show sfGFP fluorescence with variability in expression but do not follow the same trend as mCherry fluorescence. Scale bars represent 1000 μm. (B) mCherry protein concentration 10 h after injection with mCherry mRNA combined with different mimiRs. Measured by ELISA assay. Significance is shown for pairwise comparisons (one-way ANOVA with Tukey's HSD post hoc test, $n = 4$ biological replicates). P values are: positive control–partial match 0.4923792, negative control–partial match 0.0111299, negative control–positive control 0.0689794. (C) Average fluorescence intensity of mCherry normalized to GFP 10 h after injection with mCherry mRNA combined with different mimiRs and sfGFP mRNA as intensity control. Significance is shown for pairwise comparisons (one-way ANOVA with Tukey's HSD post hoc test, $n = 3$ biological replicates). P values are: positive control–partial match 0.1583761, negative control–partial match 0.0116010, negative control–positive control 0.1534842. Data information: in (B, C) box plots show median, the lower and upper bounds correspond to the 25th and 75th percentiles and whiskers extend to maximum and minimum values. The statistically significant difference is represented by: *P value < 0.05, **P value < 0.01, ***P value < 0.001, ns not significant. Source data are available online for this figure.

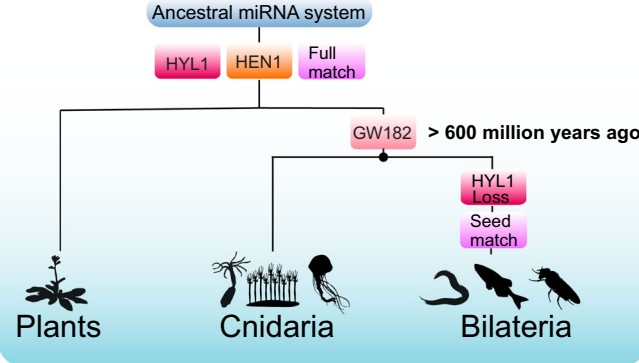

**Figure 6.  Simplified schematics highlighting similarities and differences in the miRNA system in plants, Cnidaria, and Bilateria.**

Results presented here and in previous publications (Moran et al, 2014, 2017; Tripathi et al, 2022; Mauri et al, 2017; Modepalli et al, 2018) could suggest that miRNA post-transcriptional regulation evolved before the separation of plants and animals. HYL1 and HEN1 were present in the common ancestor of plans and animals where they had roles in miRNA biogenesis and methylation of miRNAs to protect from degradation, respectively. GW182 was mitigating target translational inhibition before the separation of Cnidaria and Bilateria over 600 million years ago. While plant and cnidarian miRNAs match their targets with nearly-full complementarity, bilaterian miRNAs evolved to depend on seed match. In addition, bilaterians lost HYL1 that was replaced with other miRNA biogenesis proteins. Animal silhouettes are from http://phylopic.org/.

## Nematostella culture and microinjection

*Nematostella* (Rhode River, Maryland, source) polyps culturing, spawning, and fertilization were conducted as previously described (Genikhovich and Technau, 2009) with minor modifications. Cultured anemones were maintained at 18 °C under dark conditions and fed with freshly hatched *Artemia salina* nauplii three times a week. Anemones were induced to release gametes in 25 °C for 8 h followed by fertilization of WT eggs with either WT or heterozygote *TBP::mCherry* sperm. The gelatinous sack surrounding the eggs was removed by incubation in 3% L-Cysteine (Merck Millipore, USA) while rotated by hand for 15 min. Microinjection to zygotes was performed with Eclipse Ti-S Inverted Research Microscopes (Nikon, Japan) connected to an Intensilight fiber fluorescent illumination system (Nikon) for visualization of the fluorescent injected mixture. The system is mounted with a NT88-V3 Micromanipulator Systems (Narishige, Japan). Every replicate

included injection of three groups of 400–700 zygotes each: negative control shRNA group, positive control mimiR, and altered mimiR. *TBP::mCherry* heterozygotes were injected with shRNA/mimiRs at 31.7 μM. WT zygotes were injected with mCherry mRNA at 0.167 μM along with shRNA/mimiR at 1 μM, and 100 mM KCl. For NveAGO IP, ~4200 pooled WT zygotes were injected with a mix of three mimiRs at 31.7 μM each. All injection mixes included dextran Alexa Fluor 4.88 (Thermo Fisher Scientific, USA) for tracing of injection mix. The injected animals were kept in an incubator at 22 °C, counted, and transferred to fresh *Nematostella* medium (16‰ artificial seawater made from dry Red Sea salt) every day. The animals were visualized before flash-frozen in liquid nitrogen. The frozen samples were kept in −80 °C until either RNA or protein extraction (~150 animals in each sample) or NveAGO IP (~1400 animals in each sample).

## Hydractinia cultures and microinjection

Adult WT *Hydractinia symbiolongicarpus* colonies were maintained as previously described (Frank et al, 2020). The colonies were grown in artificial seawater at 19–22 °C on glass slides, separated to males and females. The animals were fed with *Artemia* nauplii four times per week, and once a week with ground oysters. The animals were kept in a constant 14:10 light:dark cycle, where females and males spawn 1.5 h after exposure to light. Zygotes were injected within 1.5 h from fertilization as previously described (Millane et al, 2011; Salinas-Saavedra et al, 2023), with GFP or mCherry mRNA at 0.167 μM and shRNA/mimiRs (Negative control shRNA, positive control mimiR or seed match mimiR) at 15.85 μM. All injection mixes included dextran Alexa Fluor 594 for tracing of injection mix and normalizing measured GFP fluorescence levels. Injected embryos were kept at room temperature, and dishwater was changed twice, until fluorescence was measured at 24 h post injection.

## Zebrafish embryos culture and microinjection

Wild-type zebrafish (AB/TL) maintenance was according to standard procedures. Fertilized eggs were collected at 28 °C and kept in culture medium (5 mM NaCl, 0.17 mM KCl, 0.33 mM $CaCl_2$, 0.33 mM $MgSO_4$, 0.25 mM HEPES, 0.1% Methylene blue). A total of ~150 embryos per group were microinjected at the one-cell stage with 1 nl of solution containing 0.297 μM mCherry mRNA, 0.3 μM sfGFP mRNA and shRNA/mimiR at 1.782 μM (Negative control shRNA, positive control mimiR or seed match mimiR). All

**Table 2. Primers and templates used in this study.**

| shRNA/mimiR templates | |
|---|---|
| Negative control—no match | AAGCAACACGCAGAGTCGTAATCTCTTGAATTACGACTCTGCGTGTTGCTATAGTGAGTCGTATTA |
| Positive control—full match | TCGCCCTGAACCTGAAACATATGACAACCATAGGTTTCAGCGTCAGGGCTATAGTGAGTCGTATTA |
| Seed match | ATGGAGATGCGGAGAAACATATGACAACCATAGGTTTCTCGCCATCTCCTATAGTGAGTCGTATTA |
| Seed + supplemental matches | ATGGACTGACGGAGAAACATATGACAACCATAGGTTTCTCGCTCAGTCCTATAGTGAGTCGTATTA |
| Mismatch positions 10-11 | TCGCCCTGAAGGTGAAACATATGACAACCATAGGTTTCACGGTCAGGGCTATAGTGAGTCGTATTA |
| Mismatch position 10 | TCGCCCTGAACGTGAAACATATGACAACCATAGGTTTCACCGTCAGGGCTATAGTGAGTCGTATTA |
| Mismatch position 11 | TCGCCCTGAAGCTGAAACATATGACAACCATAGGTTTCAGGGTCAGGGCTATAGTGAGTCGTATTA |
| Positive control for mRNA | GCGTCACAAATTTCACAAATATGACAACCATAGTTGTGAAGGTTGTGACTATAGTGAGTCGTATTA |
| **qPCR primers** | |
| mChery Forward | GACATCCTGTCCCCTCAGTTC |
| mCherry Reverse | GGGGAAGGACAGCTTCAAGTA |
| HKG4 Forward | GCTCAAACCTGGTCTTCTACCTATG |
| HKG4 Reverse | GCGATGGGTGCAATGACA |
| **mRNA primers** Nematostella | |
| Cloning Forward | TGTTAAACCAACCAACCACCATGGTG |
| Cloning Reverse | GCAGTGAAAAAAATGCTTCTATTTGTG |
| 3' RACE PCR | CAAGTTGGACATCACCTCCCACAACG |
| 3' RACE nested PCR | ACTACACCATCGTGGAACAGTACGAAC |
| **mRNA primers** Nematostella **and** Hydractinia | |
| PCR Forward + T7 class II phi2.5 promoter | TTTAATACGACTCACTATTAGGTGTTAA |
| PCR Reverse | GCAGTGAAAAAAATGCTTCTATTTGTG |
| **mRNA primers zebrafish** | |
| PCR Forward | TAATACGACTCACTATAAGTGTTAAACCAA |
| PCR Reverse | GCAGTGAAAAAAATGCTTCTATTTGTGAAA |
| **mRNA template** | |
| mCherry mRNA Nematostella and zebrafish | TAATACGACTCACTATAGTGTTAAACCAACCAACCACCATGGTGAGCAAGGGCGAGGAGGATAACATGGCCATC ATCAAGGAGTTCATGCGCTTCAAGGTGCACATGGAGGGCTCCGTGAACGCCCACGAGTTCGAGATCGAGGGCGA GGGCGAGGGCCGCCCCTACGAGGGCACCCAGACCGCCAAGCTGAAGGTGACCAAGGGTGGCCCCCTGCCCTTCG CCTGGGACATCCTGTCCCCTCAGTTCATGTACGGCTCCAAGGCCTACGTGAAGCACCCCGCCGACATCCCCGAC TACTTGAAGCTGTCCTTCCCCGAGGGCTTCAAGTGGGAGCGCGTGATGAACTTCGAGGACGGCGGCGTGGTGAC CGTGACCCAGGACTCCTCCCTCCAGGACGGCGAGTTCATCTACAAGGTGAAGCTGCGCGGCACCAACTTCCCCT CCGACGGCCCCGTAATGCAGAAGAAGACCATGGGCTGGGAGGCCTCCTCCGAGCGGATGTACCCCGAGGACGGC GCCCTGAAGGGCGAGATCAAGCAGAGGCTGAAGCTGAAGGACGGCGGCCACTACGACGCTGAGGTCAAGACCAC CTACAAGGCCAAGAAGCCCGTGCAGCTGCCCGGCGCCTACAACGTCAACATCAAGTTGGACATCACCTCCCACA ACGAGGACTACACCATCGTGGAACAGTACGAACGCGCCGAGGGCCGCCACTCCACCGGCGGCATGGACGAGCTG TACAAGTAAGACTCTAGATCATAATCAGCCATACCACATTTGTAGAGGTTTTACTTGCTTTAAAAAAACCTGAAA CATACCCACACCTCCCCCTGAACCTGAAACATAAATGAATGCAATTGTTGTTGGAAACATATTAACTTGTTTA TTGCAGCTTATAATGGTTACAAATGAAAGCAATAGCATCACAAATTTCACAAATAGAAGCATTTTTTTCACTGC |
| mCherry mRNA zebrafish (adjustment for transcription kit in bold) | TAATACGACTCACTATAAGTGTTAAACCAACCAACCACCATGGTGAGCAAGGGCGAGGAGGATAACATGG CCATCATCAAGGAGTTCATGCGCTTCAAGGTGCACATGGAGGGCTCCGTGAACGGCCACGAGTTCGAGATCG AGGGCGAGGGCGAGGGCCGCCCCTACGAGGGCACCCAGACCGCCAAGCTGAAGGTGACCAAGGGTGGCCCCC TGCCCTTCGCCTGGGACATCCTGTCCCCTCAGTTCATGTACGGCTCCAAGGCCTACGTGAAGCACCCCGCC GACATCCCCGACTACTTGAAGCTGTCCTTCCCCGAGGGCTTCAAGTGGGAGCGCGTGATGAACTTCGAGGA CGGCGGCGTGGTGACCGTGACCCAGGACTCCTCCCTCCAGGACGGCGAGTTCATCTACAAGGTGAAGCTGCG CGGCACCAACTTCCCCTCCGACGGCCCCGTAATGCAGAAGAAGACCATGGGCTGGGAGGCCTCCTCCGAGCGGA TGTACCCCGAGGACGGCGCCCTGAAGGGCGAGATCAAGCAGAGGCTGAAGCTGAAGGACGGCGGCCACTACGA CGCTGAGGTCAAGACCACCTACAAGGCCAAGAAGCCCGTGCAGCTGCCCGGCGCCTACAACGTCAACATCAAGT TGGACATCACCTCCCACAACGAGGACTACACCATCGTGGAACAGTACGAACGCGCCGAGGGCCGCCACTCCACCG GCGGCATGGACGAGCTGTACAAGTAAGACTCTAGATCATAATCAGCCATACCACATTTGTAGAGGTTTTAC TTGCTTTAAAAAAACCTGAAACATACCCACACCTCCCCCTGAACCTGAAACATAAATGAATGCAATTGTTGTT GGAAACATATTAACTTGTTTATTGCAGCTTATAATGGTTACAAATGAAAGCAATAGCATCACAAATTTCACAAATAG AAGCATTTTTTTCACTGC |

**Table 2.** (continued)

| sfGFP mRNA zebrafish | AATACACTTGTTCTTTTTGCAATATTCAAGCTCATCGATTCGAATTCATGCCTAAGAAGAAGAGAAAGGTGG<br>TGTCTAAAGGAGAGGAGCTGTTCACAGGCGTGGTGCCAATCCTGGTGGAGCTGGATGGAGACGTGAACGGCCA<br>CAAGTTCAGCGTGAGAGGCGAGGGAGAGGGAGACGCCACAAACGGCAAGCTGACACTGAAATTCATCTGCAC<br>AACAGGCAAACTGCCTGTGCCTTGGCCAACCCTGGTGACAACCCTGACATACGGAGTGCAGTGCTTTAGCAGA<br>TACCCTGATCACATGAAACAGCACGATTTCTTCAAGAGCGCCATGCCTGAGGGCTACGTGCAGGAGAGAACCA<br>TCAGCTTCAAGGATGACGGAACCTACAAGACAAGAGCCGAGGTGAAGTTTGAGGGAGATACACTGGTGAACA<br>GAATCGAGCTGAAAGGCATCGATTTCAAAGAGGATGGCAACATCCTGGGACACAAACTGGAGTACAACTTCAACA<br>GCCACAACGTGTACATCACAGCCGATAAACAGAAGAACGGCATCAAAGCCAACTTCAAGATCAGACACAATGTG<br>GAGGATGGATCTGTGCAGCTGGCCGATCACTACCAGCAGAACACACCTATCGGAGACGGCCCAGTGCTGCT<br>GCCAGATAACCACTACCTGAGCACACAGAGCGTGCTGTCTAAAGACCCTAACGAGAAGAGAGATCACAT<br>GGTGCTGCTGGAGTTTGTGACAGCCGCCGGAATCACCCTGGGCATGGATGAGCTGTACAAACCTGCTGCTAAG<br>AGAGTGAAACTGGATTAGCTCGAGGATGCTAGGAGATCTGAGTTCAAGGATCCTGATTGTGGTAGTGATCTGC<br>CTTTCTTTCTTTTTTTTTTTTTTGCCTGATCATCTCACACCCTTTCTTTCTTTTTTTTTTTTTGCCTGATCCATGAC<br>TCCTGTGATGGTATCTAGAACTATAGTGAGTCGTATTACACTAGTAAAAAAAAAAAAAAAAAAAAAAAAAAAAAA<br>AAAAAAAAAA |
| eGFP mRNA *Hydractinia* | TAATACGACTCACTATTAGGTGTTAAACCAACCAACCACCATGGTATCCAAGGGCGAAGAATTATTTACTGGTG<br>TTGTACCCATATTGGTGGAGCTCGACGGAGATGTAAATGGACACAAGTTCAGTGTGTCTGGGGAAGGAGAAGGAG<br>ATGCCACCTATGGAAAGCTGACTTTAAAGTTCATCTGTACTACGGGCAAGTTGCCTGTTCCTTGGCCTACACTTGT<br>CACAACTCTGACATATGGTGTTCAATGCTTTTCGCGGTATCCGGATCATATGAAGCAGCACGATTTTTTTAAGAGTGC<br>GATGCCAGAAGGTTATGTTCAGGAAAGGACCATATTTTTCAAGGACGATGGAAATTATAAAACCAGAGCTGAGGTAA<br>AATTTGAAGGAGATACATTGGTTAATCGCATTGAATTAAAAGGAATCGATTTTAAAGAGGATGGTAACATCCTCGGT<br>CATAAACTTGAGTATAACTACAACTCACATAACGTCTACATAATGGCTGATAAACAAAAAAATGGCATTAAAGTCAA<br>CTTTAAAATACGTCATAACATTGAAGACGGTTCAGTCCAACTTGCCGATCACTATCAACAAAACACTCCTATTGGTGACG<br>GTCCAGTTTTGTTACCAGACAACCACTACCTATCTACACAAAGCGCCTTAAGCAAAGACCCAAATGAAAAAAGAG<br>ATCATATGGTTTTGCTGGAATTTGTTACAGCAGCTGGAATTACACTAGGAATGGATGAATTATACAAATAAGA<br>CTCTAGATCATAATCAGCCATACCACATTTGTAGAGGTTTTACTTGCTTTAAAAAACCTGAAACATACCCAC<br>ACCTCCCCCTGAACCTGAAACATAAAATGAATGCAATTGTTGTTGGAAACATATTAACTTGTTTATT<br>GCAGCTTATAATGGTTACAAATGAAAGCAATAGCATCACAAATTTCACAAATAGAAGCATTTTTTTCACTGC |
| mCherry mRNA *Hydractinia* | TAATACGACTCACTATTAGGTGTTAAACCAACCAACCACCATGGTCTCGAAAGGAGAAGAGGATAATAT<br>GGCTATAATTAAGGAATTTATGCGCTTTAAAGTGCACATGGAGGGCTCCGTGAACGGACACGAATTTGA<br>AATTGAAGGCGAAGGTGAGGGAAGACCATACGAAGGAACACAGACCGCTAAACTTAAGGTCACAAAGG<br>GCGGTCCATTACCATTTGCATGGGACATCCTTTCACCCCAGTTCATGTACGGAAGCAAGGCATACGTTA<br>AACATCCCGCTGACATCCCCGACTACTTGAAGTTGAGCTTTCCAGAGGGATTTAAGTGGGAAAGAGTG<br>ATGAATTTCGAGGATGGAGGGGTTGTTACGGTAACCCAAGATTCCAGTCTCCAGGACGGGGAATTTAT<br>CTACAAGGTAAGCTAAGAGGTACAAATTTTCCGAGTGATGGGCCTGTTATGCAAAAGAAGACAATGG<br>GTTGGGAAGCCAGCTCCGAGCGCATGTACCCCGAAGACGGAGCTTTAAAAGGAGAAATCAAGCAGCGA<br>CTGAAATTGAAAGATGGAGGTCATTATGATGCAGAAGTTAAAACTACTTACAAAGCCAAAAAACCAGT<br>TCAGTTACCAGGTGCATATAATGTTAACATAAAACTAGACATCACATCACACAACGAGGACTATACGATC<br>GTTGAACAATACGAGCGCGCGGAGGGTAGACATTCTACCGGTGGAATGGACGAGCTATATAAATAAGA<br>CTCTAGATCATAATCAGCCATACCACATTTGTAGAGGTTTTACTTGCTTTAAAAAACCTGAAACATACCCAC<br>ACCTCCCCCTGAACCTGAAACATAAAATGAATGCAATTGTTGTTGGAAACATATTAACTTGTTTATTGCAGC<br>TTATAATGGTTACAAATGAAAGCAATAGCATCACAAATTTCACAAATAGAAGCATTTTTTTCACTGC |

DNA sequences used in this study.

injection mixes included 10% phenol red (New England Biolabs) for tracing of injection mix. Zebrafish embryos were visualized under a fluorescent stereomicroscope before dechorionated and frozen at 10 h post injection. For removal of chorion, embryos were incubated for 5 min with 1 mg/ml Pronase (Merck Millipore) then washed with culture medium and flash-frozen in liquid nitrogen. The frozen samples were kept in −80 °C until protein extraction. All protocols and procedures involving zebrafish were approved by the Institutional Committee on Animal Care and Use (IACUC, Protocol #NS-15859), The Alexander Silverman Institute of Life Sciences, The Hebrew University of Jerusalem.

## RNA extraction

Total RNA was extracted from ~150 injected animals (3 days old planulae) with the aid of Tri-Reagent® (Merck Millipore) according to the manufacturer's protocol, with a few minor changes. At the RNA isolation phase, samples were centrifuged at 21,130 × *g*. Removal of residual genomic DNA from the extracted RNA was conducted by treatment with Turbo DNase twice for 30 min at 37 °C (Thermo Fisher Scientific) and repeating the RNA purification procedure for a second time. Final RNA pellets were resuspended in 23–25 μl of RNase-free water (Merck Millipore). Final concentration was measured by Qubit™ RNA BR Assay Kit (Thermo Fisher Scientific). RNA integrity was assessed by gel electrophoresis with 1:1 formamide (Merck Millipore) and 1 μl of loading dye on 1.5% agarose gel. RNA samples were stored at −80 °C until used. For sRNA libraries preparation, RNA was extracted from immunoprecipitated samples using only one round of RNA purification, without DNase treatment. In total, 1 μl glycogen (Merck Millipore) was added at the RNA isolation step. RNA pellets were washed twice with 75% ethanol and resuspended in 8 μl of RNase-free water.

## shRNA/mimiR design

shRNA sequence to serve as negative control with no matches in *Nematostella* genome was taken from an existing protocol (Karabulut et al, 2019). mimiRs to target mCherry transcript were designed based on *Nematostella* miR-2022, an endogenous miRNA

stem-loop that was used as template to allow better prediction of cleaving sites by Dicer and to ensure selection of the desired strand and loading onto NveAGO1 (Fridrich et al, 2020; Moran et al, 2014). The targeted sequence was selected have U as a 5' terminal nucleotide, according to *Nematostella* guide strand characteristics, and mismatches were introduced to the predicted star strand in positions 1, 8, 9, and 17 (Fridrich et al, 2020). mimiRs were designed to the 3' UTR region of the mCherry transcript. The base in position 19 was always cytosine, due to in vitro transcription requirements. mimiR sequence alterations included mismatches in positions 10–11, position 10 or 11, only positions 1–8 base-pairing with the mCherry sequence (seed match), and positions 1–8 and 13–16 matching (seed + supplemental matches).

## In vitro transcription

The shRNA and mimiRs were transcribed according to the manufacturer's instructions using the AmpliScribe™ T7-Flash™ Transcription kit protocol (Lucigen, USA) with a few changes. The shRNA/mimiR DNA templates were ordered from Integrated DNA Technologies, Inc (Integrated DNA Technologies, USA) as reverse complements to the sequence of T7 promoter followed by shRNA/mimiR precursor. Templates were annealed with a T7 promoter primer prior to the in vitro transcription reaction, which was carried out for 15 h, followed by addition of 1 μl of DNase, incubation at 37 °C for 15 min, and product cleaned up with Quick-RNA MiniPrep Kit (Zymo Research, USA). The concentration of transcripts was measured by Epoch Microplate Spectrophotometer (BioTek Instruments, Cole-Parmer, USA) and product integrity was validated with gel electrophoresis. The ready to use hairpins were kept at −80 °C until used. All shRNA and mimiR templates used in this study appear in Table 2.

## cDNA synthesis

cDNA synthesis was conducted with iScript™ (Bio-Rad, USA) according to the manufacturer's protocol. 100 ng of RNA (extracted from ~150 treated 3-day-old planulae) per sample was used as a template, resulting in final concentration of 5 ng/μl. cDNA was stored at −20 °C.

## Reverse transcription-quantitative PCR

Primers to amplify mCherry transcript for RT-qPCR were designed via Primer3 version 0.4.0 (Untergasser et al, 2012) and calibrated at concentrations of 25, 5, 1, 0.2, and 0.04 ng/μl to generate standard curves with StepOnePlus Real-Time PCR System v2.2 (ABI, Thermo Fisher Scientific). Primer quality was 125% efficiency, −2.83 slope and $R^2 > 0.99$. The specificity of the amplified products was determined by the presence of a single peak in the melting curve. RT-qPCR was performed using StepOnePlus Real-Time PCR System v2.2 (ABI, Thermo Fisher Scientific) and cDNA amplification was quantitatively assessed with using Fast SYBR Green Master Mix (Thermo Fisher Scientific). Each sample was quantified in triplicates for mCherry transcript and housekeeping gene 4 (HKG4) as an internal control (Columbus-Shenkar et al, 2018). Overall, 1 μl of cDNA template was used for all replicates. For negative control cDNA was replaced with RNase-free water. The Reaction thermal profile was 95 °C for 20 s, then 40 amplification cycles of 95 °C for

3 s and 60 °C for 30 s, a dissociation cycle of 95 °C for 15 s and 60 °C for 1 min and then brought back to 95 °C for 15 s ( + 0.6 °C steps). mCherry fold change was analyzed using a comparative Ct method ($2^{-\Delta\Delta Ct}$) (Schmittgen and Livak, 2008). Thresholds for HKG and mCherry detection were equalized between individual experiments. Each experiment was composed of at least three biological replicates. All primers used for qPCR analysis are listed in Table 2.

## Protein extraction

Total protein extraction was implemented by adding 200 μl of the following lysis buffer: 50 mM Tris-HCl (pH 7.4), 150 mM KCl, 10% glycerol, 0.5% NP-40, 5 mM EDTA (all chemicals purchased from Merck Millipore) and Halt™ Protease Inhibitor cocktail (Thermo Fisher Scientific). Then, samples were homogenized with pestle mixer (Argos Technologies, cat. No. A0001) and incubated at 4 °C for 2 h in rotating mixer (Intelli Mixer™ RM-2, ELMI, function 1, 7 rpm). Samples were later centrifuged at $16,000 \times g$ for 15 min at 4 °C and the aqueous phase was transferred to a new tube. The concentration of total protein was measured using Pierce™ BCA Protein Assay Kit (Thermo Fisher Scientific) on Epoch Microplate Spectrophotometer (BioTek Instruments). Samples were kept at −80 °C until used.

## Red fluorescence protein enzyme-linked immunosorbent assay (RFP ELISA)

mCherry protein levels were detected with the aid of RFP ELISA kit (Cell Biolabs, Inc., USA). All protein samples were diluted to equal concentration prior to loading on antibody plate, and the experiment was carried out according to protocol. Epoch Microplate Spectrophotometer (BioTek Instruments) was used for absorbance measuring. Fit for standard curve was found using CurveExpert Basic 2.2.3 (Hyams Development, USA).

## Rapid amplification of cDNA ends (3' RACE)

In order to reveal the exact length of the 3' end of the transgenic *TBP::mCherry* transcript, SMARTer® RACE 5'/3' Kit was used (Takara Bio, Japan). Prior to the reaction, RNA was extracted from 3-month-old animals, with one round of Tri-Reagent® (Merck Millipore). cDNA synthesis was conducted according to the manufacturer's protocol with 500 ng of RNA. Gene specific primers were designed for the PCR and Nested PCR reactions, which were carried out by Advantage® 2 Polymerase (Takara Bio). Final products were outsourced for Sanger sequencing (HyLabs, Israel). Primers used for 3' RACE are listed in Table 2.

## mCherry and GFP mRNA generation

mRNA templates were ordered as gBlock gene fragments (Integrated DNA Technologies). The sequences included T7 promoter, EF1α kozak sequence TGTTAAACCAACCAACCACC and 3' UTR with three seed sites 21 bases apart (two were inserted in addition to the original one). In addition, the 3' UTR included one site for full match mimiR and two nucleotides' changes to make gBlock synthesis efficient. Codon-optimized GFP and mCherry mRNA sequences were designed for expression in *Hydractinia*. The DNA fragment was dissolved in TE buffer to final 20 ng/μl and incubated at 50 °C for 20 min. For injection to *Nematostella*, the

template was cloned to pGEM®-T Easy plasmid (Promega) and amplified with forward primer to add T7 promoter class II phi2.5. In vitro transcription was conducted with HighYield T7 Cap 1 AG (3'-OMe) mRNA Synthesis Kit (m5CTP) (Jena Bioscience, Germany) using 800 ng of amplified template followed by Turbo DNase treatment (Thermo Fisher Scientific) by incubation with 1 µl of DNase for 30 min at 37 °C twice sequentially. For injection to *Hydractinia* and zebrafish the mRNA template was amplified from gBlock, zebrafish mRNA was amplified with 68 °C annealing temperature. mRNA was transcribed with HiScribe T7 mRNA Kit with CleanCap Reagent AG (New England Biolabs) according to the manufacturer's protocol with 1 µg of amplified template. In vitro transcription products were cleaned using RNA clean and concentrator 25 (Zymo Research) and eluted with 33 µl RNase-free water. Concentration was measured using Qubit™ RNA Broad Range Assay Kit with the Qubit Fluorometer (Thermo Fisher Scientific). Poly-adenylation followed using *Escherichia coli* Poly(A) Polymerase (New England Biolabs) for 30 min at 37 °C and products were further cleaned with RNA clean & concentrator 5 (Zymo Research) and eluted with 8–10 µl RNase-free water. Single product was validated on 1.5% agarose gel after incubated at 95 °C for 2 min in thermo cycler with hot lid then brought to 22 °C and mixed with formamide (Merck millipore) in 1:3 ratio. The mRNA was stored in −80 °C until injected. mRNA templates and primers used for cloning and amplification appear in Table 2.

## sfGFP mRNA generation

sfGFP-encoding mRNA with a 40 nucleotides polyA tail was in vitro transcribed from a plasmid encoding the construct under SP6 promoter. The plasmid was linearized by digestion with SapI restriction enzyme (New England Biolabs) for 1 h in 37 °C. mRNA was synthesized using HiScribe SP6 RNA synthesis kit (New England Biolabs) according to the the manufacture's protocol. Resulting mRNA levels were quantified by nanodrop (NanoDrop Microvolume Spectrophotometer, Thermo Fisher Scientific), and mRNA length was validated by gel electrophoresis. The mRNA was stored in −80 °C until injected.

## NveAGO Immunoprecipitation

NveAGO1 and NveAGO2 IP were performed as previously described with custom antibodies (GenScript, USA) and rabbit IgG antibody as control group (Merck Millipore, I5006-10MG) (Fridrich et al, 2020). In brief, 100 µl of Protein A Magnetic Beads (MedChem Express, USA) were washed five times in 1 ml of 1× PBS. Then, 5 µg antibodies were added with 1.3 ml 1× PBS and incubated overnight at 4 °C in rotation. Frozen animal samples were treated and homogenized with 1 ml of the following lysis buffer: 25 mM Tris-HCl (pH 7.4), 150 mM KCl, 25 mM EDTA, 0.5% NP-40, 1 mM DTT, Protease inhibitor cOmplete ULTRA tablets (Roche), Protease Inhibitor Cocktail Set III, EDTA-Free (Merck Millipore) and Murine RNAse inhibitor (New England Biolabs). The samples were then rotated for 2 h in 4 °C followed by centrifugation at 16,000 × g for 15 min at 4 °C and the aqueous phase was collected. To preclear the lysate, 100 µl of beads were washed three times in 1 ml of 1× PBS and one time in lysis buffer then the lysate was added to the washed beads. Lysis buffer was added to complete 1.2 ml and the samples were rotated at 4 °C for

an hour. Next, the pre-cleared lysate of each biological replicate was split into three and added to the antibody-bound beads. The samples were then incubated for 2 h in rotation at 4 °C followed by removal of the lysate and washing of the beads five times with the following wash buffer: 50 mM Tris-HCl (pH 7.4), 300 mM NaCl, 5 mM MgCl₂ 0.05% NP-40, Protease inhibitor cOmplete ULTRA tablets (Roche), Protease Inhibitor Cocktail Set III, EDTA-Free (Merck Millipore) and Murine RNAse inhibitor (New England Biolabs). The immunoprecipitated samples were kept in −80 °C until RNA extraction.

## sRNA libraries

RNA extracted from samples after NveAGO IP was size-selected on 15% urea-PAGE gel (Bio-Rad). RNA in the size of 18–30 nucleotides was cut from the gel and extracted by incubation in 810 µl 0.3 M NaCl overnight at 4 °C in rotation. Size-selected sRNAs were precipitated at −20 °C for 3 h in 1 ml ice-cold 100% ethanol with 1 µl glycogen (Merck Millipore), then with 900 µl of ice-cold 75% ethanol, with centrifugation at 21,130 × g and resuspended in 7.5 µl RNase-free water. Next, library preparation was conducted using NEBNext Multiplex Small RNA Library Prep Set for Illumina kit (New England Biolabs) with modified protocol adapted to low RNA input (Plotnikova et al, 2019). Library preparation kit adapters 3' and 5' ligated to sRNAs and reverse transcription primer were diluted 1:3 of stock concentration before use. For overnight ligation of 3' adapter, T4 RNA Ligase 2 truncated KQ and Murine RNAse inhibitor were used (New England Biolabs). For PCR amplification 18 cycles were used and amplified cDNA was run on 2% Certified™ Low Range Ultra Agarose (Bio-Rad) gel stained with orange gel loading dye (New England Biolabs). cDNA in the size of 138-150 nucleotides was cut from gel (corresponding to size-selected sRNAs) and extracted with NucleoSpin Gel and PCR Clean-up (Macherey-Nagel, Germany) with elution volume of 12 µl. Final concentrations were measured with Qubit™ dsDNA HS (high sensitivity) Assay Kit (Thermo Fisher Scientific). The quality of the libraries was determined by TapeStation system on High Sensitivity D1000 ScreenTape (Agilent Technologies, USA) before sequencing with NextSeq2000 (Illumina). The sRNA libraries included two distinct biological replicates for each NveAGO and IgG IP for determination of mimiR presence. sRNA reads were analyzed with mirDeep2 (Friedländer et al, 2012). Reads shorter than 18 nucleotides were discarded. Bowtie1 (version 1.3.1) (Langmead et al, 2009) was used to map filtered reads to *Nematostella* genome. mirDeep2 quantifier.pl module was used with default parameters for the quantification of mimiR reads and mapping of the reads to precursors. Reads mapping to mimiRs guide and star strands were normalized to the number of reads unmapped to the genome and multiplied by 1 million. The normalized reads were averaged between the two biological replicates.

## Microscopy

Fluorescence of mCherry and sfGFP protein expressed in *Nematostella* and zebrafish was detected by SMZ18 stereomicroscope (Nikon) connected to an Intensilight fiber illumination fluorescent system (Nikon). Images were captured by DS-Qi2 SLR camera (Nikon) and were analyzed and processed with NIS-Elements Imaging Software (Nikon). Zebrafish images were taken using both red and green channels and analyzed using ImageJ software (Schindelin et al, 2012). Between 17 and 29 zebrafish embryos were selected per field with three biological

replicates per treatment. For quantitative analysis, background intensity calculated by averaging the intensity of the same five locations in each field was subtracted before raw intensity was measured for each individual embryo. mCherry intensity was normalized to GFP by dividing the values. The average normalized mCherry intensity was calculated for each treatment in three biological replicates. Fluorescence of GFP and red dextran Alexa Fluor 594 in *Hydractinia* was detected and documented by Olympus SZX16 stereomicroscope with a DP71 digital mounted camera in red and green channels and analyzed in a similar manner with a few differences. Between 4 and 12 *Hydractinia* embryos were selected per field and the median pixel intensity was calculated. Embryos that presented poor health or impaired development as well as low dextran levels indicating unsuccessful injection, were excluded from the analysis. For the embryos that were included, the area around the injection point was excluded from intensity calculation due to concentrated localization of the dextran tracer, typical for *Hydractinia* injections.

## Protein structure prediction and visualization

Structures of NveAGO1 and NveAGO2 were predicted using ColabFold (Mirdita et al, 2022; Jumper et al, 2021). Predicted structures of atAGO10 and hsAGO2 were taken from Uniprot database (The UniProt Consortium, 2023). Crystal structures of atAGO10 (Xiao et al, 2023b, data ref: Protein Data Bank 7SVA, 2022) and hsAGO2 (Schirle and MacRae, 2012, data ref: Protein Data Bank 4OLA, 2014) were taken from PDB database. Structure alignments were done by RCSB PDB Pairwise Structure Alignment tool (Berman et al, 2000). Visualization of aligned structures was done using PyMOL (Schrödinger, LLC).

## Statistical analysis

Comparisons between groups of injected animals in transcript levels, protein concentration or fluorescence were tested with one-way ANOVA with Tukey's HSD post hoc test or with Welch's ANOVA with Games–Howell post hoc test without the assumption of homogeneity of variances. The normality of the data was validated beforehand. No statistical method was used to predetermine sample size. Embryos for injection were randomly collected from natural mating of the organisms in the study. No blinding was done during experiments. For mCherry transcript level, $\Delta$Ct values were compared between groups. For normalized GFP intensity, 24–31 individual embryos were compared between groups. For normalized mCherry intensity, average intensities were compared between groups. Equivalence was tested with Welch modified two-sample *t* test (TOST). All experiments that were statistically analyzed included at least three biological replicates and three technical replicates for RT-qPCR and two for ELISA. NveAGOs IP experiment included two biological replicates for determining presence of mimiR reads. The tests were performed in Rstudio 2021.09.0 (Schirle and MacRae, 2012).

## Data availability

The sRNA sequencing data from NveAGO immunoprecipitation are available on the SRA repository with the following link: https://www.ncbi.nlm.nih.gov/sra/PRJNA1111011.

The source data of this paper are collected in the following database record: biostudies:S-SCDT-10_1038-S44319-024-00350-z.

## Peer review information

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

## Acknowledgements

The authors thank Dr. Michal Bronstein and Ms. Adi Turjeman (The Center for Genomic Technologies, The Hebrew University of Jerusalem) for their help with high-throughput sequencing and Prof. Liran Carmel (Department of Genetics, The Hebrew University of Jerusalem) for providing advice on statistical analysis. This work was supported by European Research Council grants CNIDARIAMICRORNA 637456 and AntiViralEvo 863809 to YM.

## Author contributions

**Yael Admoni**: Data curation; Formal analysis; Validation; Investigation; Visualization; Methodology; Writing—original draft; Writing—review and editing. **Arie Fridrich**: Supervision; Validation; Writing—review and editing. **Paris K Weavers**: Formal analysis; Investigation; Visualization; Methodology; Writing—review and editing. **Reuven Aharoni**: Validation; Investigation; Methodology; Writing—review and editing. **Talya Razin**: Supervision; Investigation; Methodology; Writing—review and editing. **Miguel Salinas-Saavedra**: Resources; Supervision; Validation; Investigation; Methodology; Writing—review and editing. **Michal Rabani**: Conceptualization; Resources; Supervision; Validation; Investigation; Methodology; Writing—original draft; Project administration; Writing—review and editing. **Uri Frank**: Resources; Supervision; Methodology; Writing—review and editing. **Yehu Moran**: Conceptualization; Resources; Formal analysis; Supervision; Validation; Investigation; Visualization; Methodology; Writing—original draft; Project administration; Writing—review and editing.

Source data underlying figure panels in this paper may have individual authorship assigned. Where available, figure panel/source data authorship is listed in the following database record: biostudies:S-SCDT-10_1038-S44319-024-00350-z.

## Disclosure and competing interests statement

YM is an Academic Editor for *EMBO Press*. This manuscript was handled without any involvement from his side in the editorial process. The remaining authors declare no competing interests.

# Expanded View Figures

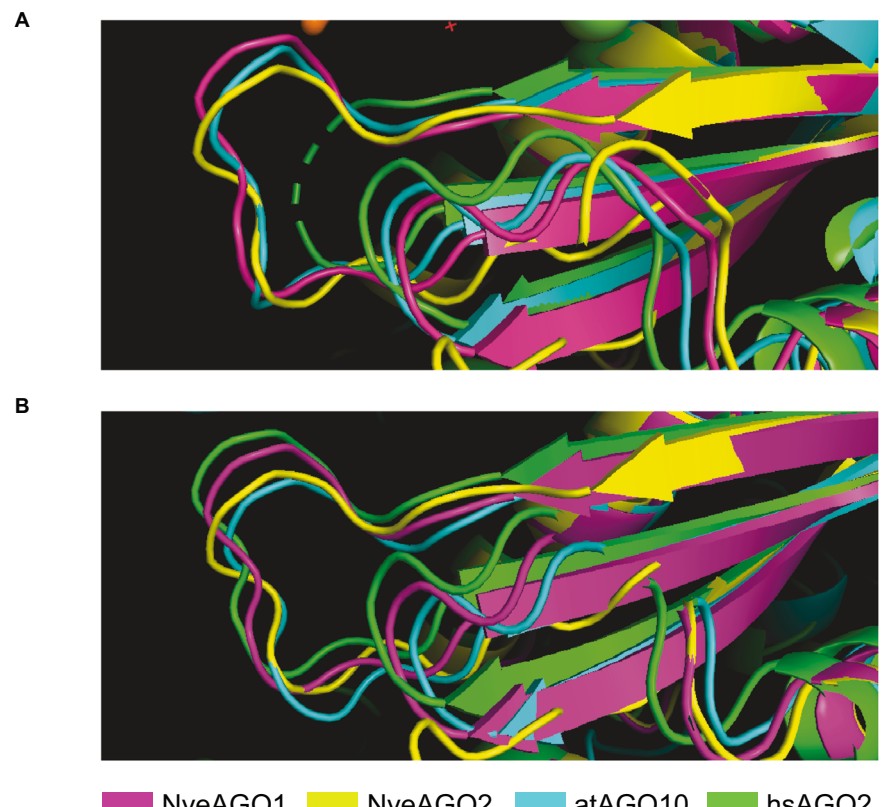

**A**

**B**

NveAGO1 NveAGO2 atAGO10 hsAGO2

**Figure EV1. Structure prediction of *Nematostella* AGOs (NveAGO) shows loop in the PIWI domain.**

(**A**) Crystalized structures of atAGO10 and hsAGO2 aligned with Colabfold predicted structures of NveAGO1 and NveAGO2. The PIWI loop appears in atAGO10, NveAGO1 and NveAGO2 but is unstructured in hsAGO2. (**B**) Aligned ColabFold predicted structures of atAGO10, hsAGO2, NveAGO1, and NveAGO2. The PIWI loop appears in all AGOs.

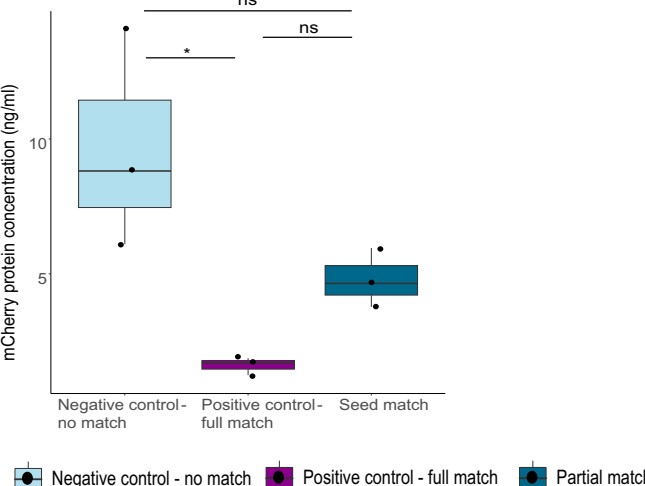

**Figure EV2.    mCherry mRNA expression in *Hydractinia* is knocked down by nearly-full match mimiR.**

mCherry protein concentration measured 24 h after injection with mCherry mRNA combined with different mimiRs to *Hydractinia* zygotes. Measured by ELISA assay. Significance is shown for pairwise comparisons (one-way ANOVA with Tukey's HSD post hoc test, $n = 3$ biological replicates). *P* values are: positive control—partial match 0.1583761, negative control—partial match 0.3124345, negative control—positive control 0.0155945. Data information: box plots show median, the lower and upper bounds correspond to the 25th and 75th percentiles and whiskers extend to maximum and minimum values. Statistically significant difference is represented by: *$P$ value < 0.05, ns not significant. Source data are available online for this figure.

**A**

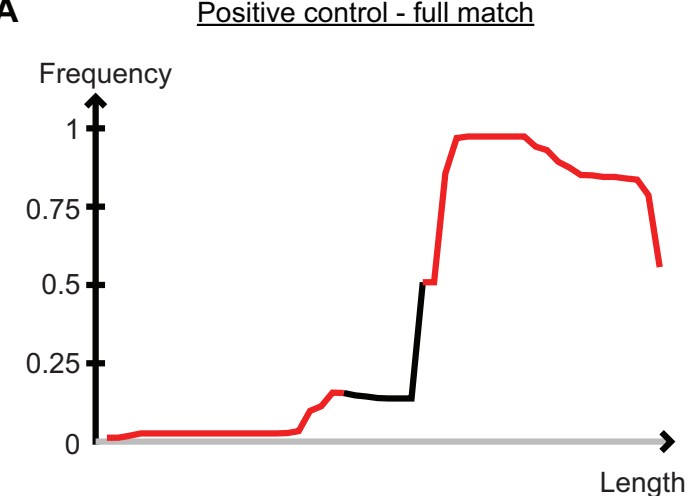

Positive control - full match

**B**

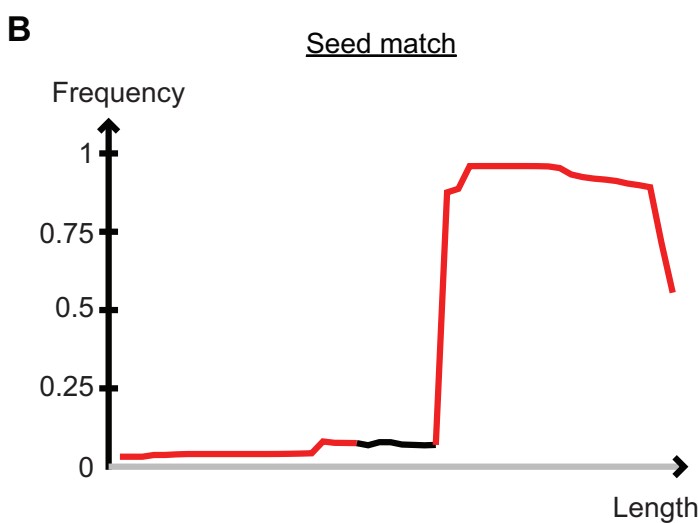

Seed match

**C**

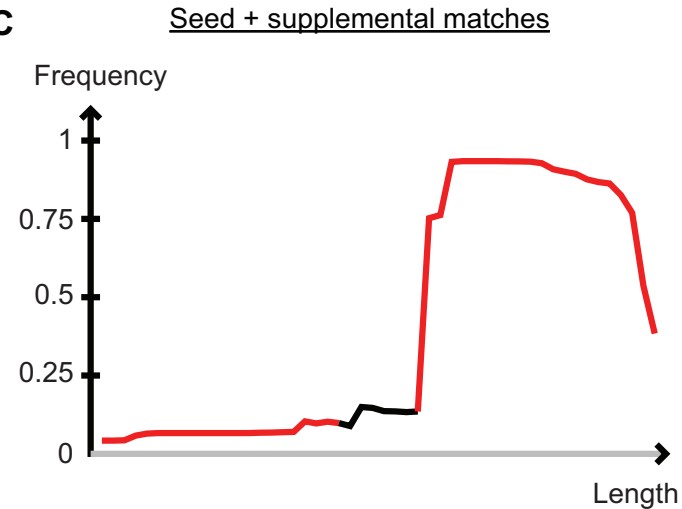

Seed + supplemental matches

**Figure EV3.   Processing of mimiRs injected to *Nematostella*.**

(**A**) Nearly-full match mimiR (positive control) reads mapped to injected precursor. (**B**) Seed match mimiR reads mapped to injected precursor. (**C**) Seed + supplemental matches mimiR reads mapped to injected precursor.

