## [Peer Review File · EMBO Reports]

miRNA-target complementarity in cnidarians resembles its counterpart in plants

Yael Admoni, Arie Fridrich, Paris Weavers, Reuven Aharoni, Talya Razin, Miguel Salinas-Saavedra, Michal Rabani, Uri Frank, and Yehu Moran

Corresponding authors: Yehu Moran (yehu.moran@mail.huji.ac.il) , Arie Fridrich (arie.fridrich@gmi.oeaw.ac.at), Yael Admoni (yael.admoni@mail.huji.ac.il)

Review Timeline:

Submission Date:	20th Nov 23
Editorial Decision:	19th Jan 24
Revision Received:	1st Jul 24
Editorial Decision:	22nd Aug 24
Revision Received:	5th Nov 24
Accepted:	21st Nov 24

Editor: *Martina Rembold*

Transaction Report:

The manuscript was processed outside of the corresponding author's access to preserve the confidentiality of the referees and independence of the editorial decision process.

19th Jan 2024

Dear Yehu,

Thank you once more for the submission of your research manuscript to EMBO Reports. I am sorry for the delay in handling it but we have now received the full set of referee reports that is copied below.

As you will see, the referees acknowledge that the findings are potentially interesting, but they also raise a number of concerns. Given that you report negative data, referee #2 considers additional controls necessary, such as the demonstration that artificial miRNAs load into the relevant Ago proteins in *Nematostella*. This control and the control experiment suggested by referee #1 should be provided. This also applies to point 2 from referee #3 on the statistical analysis. I would also encourage you to perform the Piwi loop swapping experiment suggested by referee #2, but am also happy to discuss this further. Certainly, the two recent manuscripts should be cited and discussed prominently. Both, referee #2 and #3 discuss alternative scenarios regarding the evolution of the RNAi pathway, the miRNA pathway and of Ago proteins and these scenarios should be taken into account in the interpretation and discussion of results. The common origin of miRNAs might be framed as one possibility rather than a clear conclusion.

Given these constructive comments, we would like to invite you to revise your manuscript with the understanding that the referee concerns (as detailed above and in their reports) must be fully addressed and their suggestions taken on board. Please address all referee concerns in a complete point-by-point response. Acceptance of the manuscript will depend on a positive outcome of a second round of review. It is EMBO Reports policy to allow a single round of revision only and acceptance or rejection of the manuscript will therefore depend on the completeness of your responses included in the next, final version of the manuscript.

We realize that it is difficult to revise to a specific deadline. In the interest of protecting the conceptual advance provided by the work, we recommend a revision within 3 months (April 18th). Please discuss the revision progress ahead of this time with the editor if you require more time to complete the revisions.

I am also happy to discuss the revision further via e-mail or a video call, if you wish.

*****IMPORTANT NOTE:

We perform an initial quality control of all revised manuscripts before re-review. Your manuscript will FAIL this control and the handling will be delayed IN CASE the following APPLIES:

- 1) A data availability section providing access to data deposited in public databases is missing. If you have not deposited any data, please add a sentence to the data availability section that explains that.
- 2) Your manuscript contains statistics and error bars based on $n=2$. Please use scatter blots in these cases. No statistics should be calculated if $n=2$.

When submitting your revised manuscript, please carefully review the instructions that follow below. Failure to include requested items will delay the evaluation of your revision.*****

- 1) a .docx formatted version of the manuscript text (including legends for main figures, EV figures and tables). Please make sure that the changes are highlighted to be clearly visible.
- 2) individual production quality figure files as .eps, .tif, .jpg (one file per figure). Please download our Figure Preparation Guidelines (figure preparation pdf) from our Author Guidelines pages <https://www.embopress.org/page/journal/14693178/authorguide> for more info on how to prepare your figures.

4) a complete author checklist, which you can download from our author guidelines (). Please insert information in the checklist that is also reflected in the manuscript. The completed author checklist will also be part of the RPF.

5) Please note that all corresponding authors are required to supply an ORCID ID for their name upon submission of a revised manuscript (). Please find instructions on how to link your ORCID ID to your account in our manuscript tracking system in our Author guidelines
()

6) We replaced Supplementary Information with Expanded View (EV) Figures and Tables that are collapsible/expandable online. A maximum of 5 EV Figures can be typeset. EV Figures should be cited as 'Figure EV1, Figure EV2' etc... in the text and their respective legends should be included in the main text after the legends of regular figures.

7) Please note that a Data Availability section at the end of Materials and Methods is now mandatory. In case you have no data that requires deposition in a public database, please state so instead of refereeing to the database. See also < <https://www.embopress.org/page/journal/14693178/authorguide#dataavailability>>. Please note that the Data Availability Section is restricted to new primary data that are part of this study.

Additional information on source data and instruction on how to label the files are available .

10) Figure legends and data quantification:

- the name of the statistical test used to generate error bars and P values,
- the number (n) of independent experiments (please specify technical or biological replicates) underlying each data point,
- the nature of the bars and error bars (s.d., s.e.m.)

- If the data are obtained from n {less than or equal to} 5, show the individual data points in addition to the SD or SEM.

- If the data are obtained from n {less than or equal to} 2, use scatter blots showing the individual data points.

11) Our journal encourages inclusion of *data citations in the reference list* to directly cite datasets that were re-used and obtained from public databases. Data citations in the article text are distinct from normal bibliographical citations and should directly link to the database records from which the data can be accessed. In the main text, data citations are formatted as follows: "Data ref: Smith et al, 2001" or "Data ref: NCBI Sequence Read Archive PRJNA342805, 2017". In the Reference list, data citations must be labeled with "[DATASET]". A data reference must provide the database name, accession number/identifiers and a resolvable link to the landing page from which the data can be accessed at the end of the reference.

Further instructions are available at .

12) All Materials and Methods need to be described in the main text. We would encourage you to use 'Structured Methods', our new Materials and Methods format. According to this format, the Materials and Methods section should include a Reagents and Tools Table (listing key reagents, experimental models, software and relevant equipment and including their sources and relevant identifiers) followed by a Methods and Protocols section in which we encourage the authors to describe their methods using a step-by-step protocol format with bullet points, to facilitate the adoption of the methodologies across labs. More information on how to adhere to this format as well as downloadable templates (.doc or .xls) for the Reagents and Tools Table can be found in our author guidelines: <

<https://www.embopress.org/page/journal/14693178/authorguide#manuscriptpreparation>>. An example of a Method paper with Structured Methods can be found here: .

13) As part of the EMBO publication's Transparent Editorial Process, EMBO Reports publishes online a Review Process File to accompany accepted manuscripts. This File will be published in conjunction with your paper and will include the referee reports, your point-by-point response and all pertinent correspondence relating to the manuscript.

Lise Roth, on behalf of Martina Rembold

***** Reviewer's comments *****

Referee #1 (Comments on Novelty/Model System for Author):

My notes to the author should clarify my choices above.

Referee #1 (Remarks for Author):

Summary:

The plant and animal miRNA pathways exhibit starkly different mechanisms of biogenesis and function, suggesting potential different origins. However, as a postdoc, Dr. Yehu Moran made the intriguing discovery that *Nematostella* miRNAs bind mRNA targets with high complementarity, leading to mRNA target cleavage-similar to the mechanism observed in plants rather than bilaterian animals. Since starting his independent position, Yehu's group has published a series of papers further defining the miRNA pathway in *Nematostella*. This study, a natural continuation of that work, convincingly demonstrates that the typical bilaterian seed-targeting mechanism does not function in *Nematostella* or *Hydractinia* (another distantly related cnidarian). Instead, targeting must occur through high complementarity binding. Considering that cnidarians are the sister group to bilaterians, this result, along with their previous studies, strongly supports the notion that the miRNA pathway in plants and animals originated from a common source, with subsequent divergences occurring in the bilaterian lineage that led to the seed-targeting mechanism. The work presented in this manuscript represents a significant advancement in our understanding of miRNA pathway evolution, and the experiments are well-designed and well-presented. I have some minor suggestions for improvement.

Minor Comments:

The main text should be revised to make it more transparent that the *Hydractinia* experiment in Figure 2 has a different design

as compared to the *Nematostella* experiment. Specifically, it should be clarified that the mCherry mRNA was injected rather than originating from transgene expression. This distinction is already made clear for the Zebrafish experiment in Figure 3, which follows a similar design of injecting the target mCherry mRNA. Notably, the Zebrafish experiment includes a miRNA-resistant GFP mRNA to control for variability in expression across embryos. It appears that the same control should have been applied in *Hydractinia*, although it was not. If this control had been conducted, there might have been less variability between replicates. However, I believe the point still stands even without this control, although the reader should be made aware of these caveats.

The methods section doesn't clearly explain why different statistical tests are used to analyze what are essentially the same type of data, ANOVA in Figure 1 vs. T-test in Figures 2 and 3.

Figure 1G,H: It would be helpful to mention in the legend that the RNA and protein levels were measured by qPCR and ELISA (I had to go to the methods for this info)

Typos and Suggested Word Changes:

Page 3, paragraph 1: I think you mean to say "mediated" not "mitigated" regarding the function of GW182

Page 3, third to last line: change "exhibit" to "exhibits"

Page 4, second to last line of the introduction: change "to" to "into"

Page 4, fourth line of Results and Discussion: Maybe "mimic" is a better word than "predict" here?

Referee #2 (Remarks for Author):

The study by Admoni et al. aims to address a basic question on microRNA (miRNA) function in cnidarians, the sister group of bilaterian animals: can miRNAs in cnidarians achieve target repression with only base pairing to the seed (nucleotides 2-8) as in bilaterians, or do they require more extensive base pairing, as in plants? Clear answers to this question have implications for evolution of the miRNA machinery (though not necessarily miRNAs themselves, appoint we will come back to), and are of broad interest.

The study uses the cnidarians *Nematostella vectensis* and *Hydractinia symbiolongicarpus*. The authors set up an artificial reporter system in which an mCherry-encoding mRNA carries potential miRNA target sites in the 3'-UTR. miRNA-guided repression is then measured by fluorescence intensity with a variety of different miRNAs, including extensively complementary, seed-match only, extensively complementary but centrally mismatched to varying degrees. The effect of seed-match interactions with multiple sites is also addressed. Overall, the results suggest that clear target repression is only achieved with extensively and centrally matched miRNA-target interactions, different from the situation in bilaterians and similar to the situation in plants. On this basis, the authors propose that there is a common evolutionary origin to miRNAs in plants and animals.

Although the topic is of interest and the results go some way in supporting the conclusions of the authors, there are important shortcomings, including missing controls, of the present version of the manuscript. In addition, the main conclusion on a common evolutionary origin of miRNAs is, I believe, not supported.

1. Since the essence of the results is negative observations (failure of seed-match only, or centrally mismatched, miRNAs to confer repression in cnidarians), great care needs to be taken with the controls to show that the artificial miRNAs tested actually have some function at all. Most importantly, it needs to be demonstrated directly that they load into the relevant Ago protein. If that cannot be clearly established, the results are susceptible to more trivial interpretation, such as

- the mutated miRNA is not processed correctly in cnidarians
- the mutated miRNA is not loaded into any Ago protein or into an irrelevant Ago protein.

I do appreciate the effort done by the authors to show that the miRNA variants in question confer target repression in zebrafish. That is encouraging, but it does not substitute for the need to see loading in Ago directly in *Nematostella*.

2. The report ignores the recent advances from the MacRae lab on pinpointing the structural basis for the different behaviour of Ago-miRNA complexes with respect to pairing requirements between human Ago2 and plant AGO1/AGO10. A surprisingly large part of this difference can be explained by sequence differences in an 8-amino acid loop in the Piwi domain, such that human Ago2 can be engineered to require extensive matches simply by substituting its Piwi loop by the one in *Arabidopsis* AGO10 (Xiao et al., 2023, EMBO Reports 24:e55806). The companion paper, showing the catalytically active conformation of AGO10, also gives a much more precise explanation for why central pairing is required for miRNA-guided endonucleolysis than what was previously available (Xiao et al., 2023, NSMB 30:778-784). Both of these papers are key for the present study. In particular,

given the advances shown in the EMBO Reports paper, I find it mandatory for the present study to examine whether a different evolutionary trajectory of the Piwi loop in cnidarian Ago (compared to bilaterian Agos) is also responsible for the apparent requirement for extensive complementarity of miRNA-target pairs for regulation. The study should, therefore, include an experiment in which the Piwi-loop of *Nematostella* Ago is replaced by that of human Ago2 and the effect of the different miRNA variants is measured. In case repression is observed, much of the requirement for direct observation of miRNA loading discussed in point 1 would also be dealt with.

3. I do not agree with the overall interpretation of the results as stated in the abstract: "Altogether, these results unravel striking similarities between plant and cnidarian miRNAs consolidating the evidence for common evolutionary origin of miRNAs in plants and animals."

I think that what the results do suggest is that Ago proteins in cnidarians, bilaterians and plants evolved independently to acquire different properties that can be seen, for example, in different miRNA-target pairing requirements. The miRNA machinery of course has a common evolutionary origin, as discussed by the authors in the form of the RNAi pathway to fight parasitic genetic elements. But this does not mean that miRNAs themselves were in place in a common ancestor of plants, cnidarians, and bilaterians. The fact that no single miRNA is conserved in sequence between animals and plants, represents, at least to me, a strong argument that the miRNA systems evolved independently (but from a common RNAi system).

4. Finally, I do not think that the referencing in general is well done in this paper. In many cases, papers are used to support a statement that do contain relevant information, but were by no means the first to provide the required evidence, in other cases, a too small selection of the relevant original literature is used. For example, the importance of the seed in plant miRNA-target interactions was established in the first papers on PHABULOSA repression by miR165/166 in 2003/2004 by the Martienssen and Bartel labs, and studies on other miRNA-target interactions by the Bartel lab in 2004-2006 further confirmed and generalized this notion. Similarly, the paper used to cite the introduction of the seed notion (Brennecke/Stark 2004) is certainly important and relevant, but the earlier contributions from Eric Lai (2002) and David Bartel (2002-2003) should not be omitted. Similarly, it would be relevant to include mention of the very clear biochemical proof of its importance (Ameres, Cell 2007) as well as the clear understanding of its importance from the variety of prokaryotic and eukaryotic Ago structures. I would be grateful if the authors would take considerably more care with the referencing in a revised version for this or another journal.

As a very final point, I wish to thank the authors for undertaking this work that I think is very important for refining our understanding of the small RNA regulatory systems came into existence and what is important for how they work. I am, for example, quite curious to learn what the examination of the Piwi loop in cnidarian Agos gives!

Referee #3 (Comments on Novelty/Model System for Author):

As detailed in the remarks to the authors, there is a mismatch between the statistical analyses and the conclusions drawn.

Referee #3 (Remarks for Author):

The authors investigate target recognition of cnidarian miRNA mimics. They detect repression of reporters with sites that have extensive complementarity to the miRNA, but they are unable to detect significant repression of reporters with sites that match only the seed of the miRNA. Thus seed-based targeting appears to have emerged in the bilaterian clade, after the divergence of cnidarians and bilaterians. Although this fundamental insight into miRNA evolution is of interest, the way in which the authors interpret this finding is of major concern.

Major concern

1. The results of the current paper do not provide useful information for evaluating whether plant and animal microRNAs have a common origin. The RNAi pathway (i.e., the use of siRNAs derived from Dicer processing of long dsRNA to direct AGO-catalyzed target slicing) is found broadly throughout most eukaryotic species, and thus is thought to have been present in the last eukaryotic common ancestor (LECA). By contrast, miRNAs are not found broadly throughout eukaryotes, and are instead found in only a small, polyphyletic subset of eukaryotic clades, scattered among a few of the 25 major clades the eukaryotic tree (for tree, see Burki et al., PMID: 31606140), suggesting that the extant miRNA pathways were not in the LECA but instead each emerged independently from the ancestral RNAi pathway. As expected for this independent emergence, the miRNAs in different clades have no evidence of sequence conservation. For example, the miRNAs conserved in plants have no resemblance to those conserved in animals or other eukaryotic clades. Moreover, where investigated, the miRNAs in different clades have quite different biogenesis pathways. For example, plant pri-miRNAs have very different secondary structures than animal miRNAs, and plant pri-miRNAs are processed by a single endonuclease in the nucleus, whereas animal pri-miRNAs are processed one endonuclease in the nucleus and then by a second in the cytoplasm. Thus, miRNA pathways appear to have arisen after the LECA (and after the earliest eukaryotic divergence, which separated the lineage that gave rise to plants from that that gave rise to animals; Burki, et al.). Note that the authors favor an alternative model of miRNA evolution in which miRNAs were in the common ancestor of plants and animals (i.e., in the LECA), citing the finding that miRNAs of both plants and cnidaria are often

methylated and that both use the Hyl1 co-factor for Dicer processing. However, both of these findings could also be explained by the presence of methylation and Hyl1 in an ancestral RNAi pathway, and the alternative model would require miRNAs to have been independently lost in a very large number of eukaryotic lineages (see Burki et al., all curiously not shown in Figure 4)-a far less parsimonious scenario than the conventional model of independent gain in a few lineages.

In the conventional model of miRNA evolution, each new miRNA pathway evolved from the ancestral RNAi pathway with the independent emergence of a cellular encoded short RNA hairpin that could be cleaved by Dicer to produce a guide RNA that directs the AGO-catalyzed slicing of a cellular mRNA. Each nascent miRNA pathway gradually diverged from the RNAi pathway, evolving specialized Dicer and AGO proteins. The animal pathway further diverged to evolve seed-based target repression, in addition to its ancestral AGO-catalyzed target slicing, although humans and many other animals have retained AGO-catalyzed slicing of some targets.

This paper explores the question of when seed-based target repression emerged in the animal lineage, concluding that seed-based repression emerged after the divergence of cnidarians and bilaterians. This conclusion is mistakenly used to support the alternative model of miRNA evolution in which miRNAs were in the common ancestor of plants and animals (i.e., in the LECA). However, in both the conventional model and this alternative model of miRNA evolution, bilaterian miRNAs are thought to have descended from a pathway in which targets underwent AGO-catalyzed slicing. Thus, the conclusion that seed-based repression emerged after the divergence of cnidarians cannot be used to distinguish between the two models, since both models propose that the seed-based repression descended from a pathway centered on AGO-catalyzed slicing. Therefore, the title and text should be revised to remove all claims that the results support a common origin for plant and animal miRNAs.

2. The statistical analyses do not support the primary biological claim of the paper. The paper claims that cnidarians do not perform seed-based target recognition. This conclusion is based on not seeing a statistically significant repression of targets with seed pairing. However, not being confident in a difference is not the same as being confident in no difference, especially when data are noisy. To conclude that seed pairing does not mediate repression, the authors should perform a statistical analysis that reports their confidence that seed pairing causes no repression. This is particularly important for the *Hydractinia* results.

Point-by-point responses, Admoni et al.

Editor notes:

As you will see, the referees acknowledge that the findings are potentially interesting, but they also raise a number of concerns. Given that you report negative data, referee #2 considers additional controls necessary, such as the demonstration that artificial miRNAs load into the relevant Ago proteins in *Nematostella*. This control and the control experiment suggested by referee #1 should be provided. This also applies to point 2 from referee #3 on the statistical analysis. I would also encourage you to perform the Piwi loop swapping experiment suggested by referee #2, but am also happy to discuss this further. Certainly, the two recent manuscripts should be cited and discussed prominently. Both, referee #2 and #3 discuss alternative scenarios regarding the evolution of the RNAi pathway, the miRNA pathway and of Ago proteins and these scenarios should be taken into account in the interpretation and discussion of results. The common origin of miRNAs might be framed as one possibility rather than a clear conclusion.

******* Reviewer's comments *******

Referee #1 (Comments on Novelty/Model System for Author):

My notes to the author should clarify my choices above.

Referee #1 (Remarks for Author):

Summary:

The plant and animal miRNA pathways exhibit starkly different mechanisms of biogenesis and function, suggesting potential different origins. However, as a postdoc, Dr. Yehu Moran made the intriguing discovery that *Nematostella* miRNAs bind mRNA targets with high complementarity, leading to mRNA target cleavage-similar to the mechanism observed in plants rather than bilaterian animals. Since starting his independent position, Yehu's group has published a series of papers further defining the miRNA pathway in *Nematostella*. This study, a natural continuation of that work, convincingly demonstrates that the typical bilaterian seed-targeting mechanism does not function in *Nematostella* or *Hydractinia* (another distantly related cnidarian). Instead, targeting must occur through high complementarity binding. Considering that cnidarians are the sister group to bilaterians, this result, along with their previous studies, strongly supports the notion that the miRNA pathway in plants and animals originated from a common source, with subsequent divergences occurring in the bilaterian lineage that led to the seed-targeting mechanism. The work presented in this manuscript represents a significant advancement in our understanding of miRNA pathway evolution, and the experiments are well-designed and well-presented. I have some minor suggestions for improvement.

Minor Comments:

The main text should be revised to make it more transparent that the *Hydractinia*

experiment in Figure 2 has a different design as compared to the *Nematostella* experiment. Specifically, it should be clarified that the mCherry mRNA was injected rather than originating from transgene expression. This distinction is already made clear for the Zebrafish experiment in Figure 3, which follows a similar design of injecting the target mCherry mRNA.

Answer:

We made it clear in the text describing the *Hydractinia* experiment that the mRNA expressing GFP was injected to zygotes (Please see our response to the following comment).

Notably, the Zebrafish experiment includes a miRNA-resistant GFP mRNA to control for variability in expression across embryos. It appears that the same control should have been applied in *Hydractinia*, although it was not. If this control had been conducted, there might have been less variability between replicates. However, I believe the point still stands even without this control, although the reader should be made aware of these caveats.

Answer:

Thank you for this suggestion. For technical reasons we could not inject mRNAs encoding for two fluorescent proteins in *Hydractinia*, so instead we injected a fluorescent tracer dye (Alexa Fluor 594) bound to dextran. We eventually chose to use GFP and not mCherry as it provides better results in this cnidarian species (Figure 3).

The methods section doesn't clearly explain why different statistical tests are used to analyze what are essentially the same type of data, ANOVA in Figure 1 vs. T-test in Figures 2 and 3.

Answer:

Following the referee's comments, we made changes in the statistical tests applied in this study, described in the Materials and Methods Statistical analysis section.

Figure 1G,H: It would be helpful to mention in the legend that the RNA and protein levels were measured by qPCR and ELISA (I had to go to the methods for this info)

Answer:

The methods qPCR and ELISA were mentioned the legend of Figure 1G,H according to the reviewer's request.

Typos and Suggested Word Changes:

Page 3, paragraph 1: I think you mean to say "mediated" not "mitigated" regarding the function of GW182

Page 3, third to last line: change "exhibit" to "exhibits"

Page 4, second to last line of the introduction: change "to" to "into"

Page 4, fourth line of Results and Discussion: Maybe "mimic" is a better word than "predict" here?

Answer:

We thank the reviewer for noticing these errors. We corrected the text according to the reviewer's suggestions

Referee #2 (Remarks for Author):

The study by Admoni et al. aims to address a basic question on microRNA (miRNA) function in cnidarians, the sister group of bilaterian animals: can miRNAs in cnidarians achieve target repression with only base pairing to the seed (nucleotides 2-8) as in bilaterians, or do they require more extensive base pairing, as in plants? Clear answers to this question have implications for evolution of the miRNA machinery (though not necessarily miRNAs themselves, as we will come back to), and are of broad interest.

The study uses the cnidarians *Nematostella vectensis* and *Hydractinia symbiolongicarpus*. The authors set up an artificial reporter system in which an mCherry-encoding mRNA carries potential miRNA target sites in the 3'-UTR. miRNA-guided repression is then measured by fluorescence intensity with a variety of different miRNAs, including extensively complementary, seed-match only, extensively complementary but centrally mismatched to varying degrees. The effect of seed-match interactions with multiple sites is also addressed. Overall, the results suggest that clear target repression is only achieved with extensively and centrally matched miRNA-target interactions, different from the situation in bilaterians and similar to the situation in plants. On this basis, the authors propose that there is a common evolutionary origin to miRNAs in plants and animals.

Although the topic is of interest and the results go some way in supporting the conclusions of the authors, there are important shortcomings, including missing controls, of the present version of the manuscript. In addition, the main conclusion on a common evolutionary origin of miRNAs is, I believe, not supported.

1. Since the essence of the results is negative observations (failure of seed-match only, or centrally mismatched, miRNAs to confer repression in cnidarians), great care needs to be taken with the controls to show that the artificial miRNAs tested actually have some function at all. Most importantly, it needs to be demonstrated directly that they load into the relevant Ago protein. If that cannot be clearly established, the results are susceptible to more trivial interpretation, such as

- the mutated miRNA is not processed correctly in cnidarians
- the mutated miRNA is not loaded into any Ago protein or into an irrelevant Ago protein.

I do appreciate the effort done by the authors to show that the miRNA variants in

question confer target repression in zebrafish. That is encouraging, but it does not substitute for the need to see loading in Ago directly in *Nematostella*.

Answer:

We thank the reviewer for this helpful suggestion. To show that the artificial miRNAs (mimiRs) are processed correctly and loaded into *Nematostella* AGOs we performed AGO immunoprecipitation (IP) using specific antibodies that have been used for IP before (Fridrich *et al*, 2020). We injected three of the mimiRs (nearly-full match mimiR that served as positive control in the experiments presented in Figures 2 and 3 and mimiRs seed match and seed + supplemental matches) to WT zygotes then performed AGO IP with IgG as negative control, followed by small RNA sequencing. The results of the sequencing clearly show that mature mimiRs are loaded into AGOs (Figure 2 D-F). The loading was mostly to AGO2, which is less specific than AGO1 in the loading preferences as it loads a wider range of miRNAs and siRNAs (Fridrich *et al*, 2020). Another observation is that the nearly-full match mimiR, which causes the strongest knockdown effect, was captured in lower reads than the alternative mimiRs. This was perhaps caused by somewhat less accurate processing as seen by the reads mapping to the precursor (Expanded View Figure 2), but despite the fewer reads it caused far more efficient knockdown than the bilaterian-like mimiRs. Overall, the processing of the mimiRs precursors was successful and the guide strand was loaded into AGOs noticeably more than the star strand.

2. The report ignores the recent advances from the MacRae lab on pinpointing the structural basis for the different behaviour of Ago-miRNA complexes with respect to pairing requirements between human Ago2 and plant AGO1/AGO10. A surprisingly large part of this difference can be explained by sequence differences in an 8-amino acid loop in the Piwi domain, such that human Ago2 can be engineered to require extensive matches simply by substituting its Piwi loop by the one in Arabidopsis AGO10 (Xiao *et al.*, 2023, EMBO Reports 24:e55806). The companion paper, showing the catalytically active conformation of AGO10, also gives a much more precise explanation for why central pairing is required for miRNA-guided endonucleolysis than what was previously available (Xiao *et al.*, 2023, NSMB 30:778-784). Both of these papers are key for the present study. In particular, given the advances shown in the EMBO Reports paper, I find it mandatory for the present study to examine whether a different evolutionary trajectory of the Piwi loop in cnidarian Ago (compared to bilaterian Agos) is also responsible for the apparent requirement for extensive complementarity of miRNA-target pairs for regulation. The study should, therefore, include an experiment in which the Piwi-loop of *Nematostella* Ago is replaced by that of human Ago2 and the effect of the different miRNA variants is measured. In case repression is observed, much of the requirement for direct observation of miRNA loading discussed in point 1 would also be dealt with.

Answer:

We thank the reviewer for this comment. We agree that the suggested PIWI loop swapping experiment could have important implications for our study; however, we believe that due to the immense time requirements to perform such an experiment on *Nematostella*, this experiment is beyond the scope of our paper. *Nematostella* generation time is about four months and to achieve homozygotes to a mutated AGO would take a minimum of eight months in addition to the experimental work. We would like to mention that currently *Nematostella* cell culture is not available.

We also agree that a comparison of *Nematostella* AGO1 and AGO2 to *Arabidopsis* AGOs would be very interesting and relevant for our study. Unfortunately, the structure of *Nematostella* AGOs is currently unsolved experimentally and reaching a crystalized structure is beyond the scope of our paper. To address the recent papers by Xiao et al., we investigated the predicted structure of *Nematostella* AGOs compared to atAGO10 and hsAGO2 (Expanded View Figure 1). The predicted structures of *Nematostella* AGOs reveal a loop that aligns with atAGO10 PIWI loop in contrast to the unstructured hsAGO2 loop. Unfortunately, however, when we used structure prediction to determine the structure of hsAGO2 as control, the PIWI loop appears as well in the human AGO, suggesting that in-silico structure predictions, even by powerful method such as AlphaFold2, are not 100% accurate and should be taken with caution. Thus, we currently cannot conclude that *Nematostella* AGOs have a similar loop structure as atAGO10, probably due to the small size of the loop that challenges the prediction tools.

3. I do not agree with the overall interpretation of the results as stated in the abstract: "Altogether, these results unravel striking similarities between plant and cnidarian miRNAs consolidating the evidence for common evolutionary origin of miRNAs in plants and animals."

I think that what the results do suggest is that Ago proteins in cnidarians, bilaterians and plants evolved independently to acquire different properties that can be seen, for example, in different miRNA-target pairing requirements. The miRNA machinery of course has a common evolutionary origin, as discussed by the authors in the form of the RNAi pathway to fight parasitic genetic elements. But this does not mean that miRNAs themselves were in place in a common ancestor of plants, cnidarians, and bilaterians. The fact that no single miRNA is conserved in sequence between animals and plants, represents, at least to me, a strong argument that the miRNA systems evolved independently (but from a common RNAi system).

Answer:

We thank the reviewer for the comment. The reviewer remarks that no sequence conservation of miRNAs between animals and plants indicates that the miRNA systems evolved independently. In response to that, we would like to note that conservation of miRNA sequence is uncommon even within each kingdom. In fact, in some cases no miRNAs are shared between close lineages (Moran *et al*, 2017). For example, no conserved miRNAs were found between demosponge, calcareous and homoscleromorph sponges. In addition, the demosponge *Amphimedon queenslandica* shares none of its eight miRNAs with other non-poriferan animals (Grimson *et al*, 2008; Robinson *et al*, 2013). Moreover, between cnidarians and bilaterians, only one miRNA is shared (miR-100), and it is absent from the cnidarian *Hydra magnipapillata* and all others hydras and jellyfish (Grimson *et al*, 2008; Krishna *et al*, 2013; Moran *et al*, 2014; Praher *et al*). When looking into plant miRNAs, they exhibit a remarkably fast evolution which makes sequence similarity rare among distant plant lineages (Axtell *et al*, 2007; Moran *et al*, 2017). This was also suggested to be the case in green algae, since no conserved plant miRNAs were found in the green alga *Chlamydomonas reinhardtii* (Voshall *et al*, 2017). Recently, miRNAs were identified in species of Amoebozoa, where the majority of sequences was unique and only a single miRNA was conserved between members of this group (Edelbroek *et al*, 2024). In conclusion, miRNA sequence conservation is

far from being guaranteed even when there is a common origin of the miRNA system. In light of this, we believe that lack of miRNA sequence conservation between plant and animals does not rule out a common origin for the miRNA system. Please see also our response to Referee #3.

4. Finally, I do not think that the referencing in general is well done in this paper. In many cases, papers are used to support a statement that do contain relevant information, but were by no means the first to provide the required evidence, in other cases, a too small selection of the relevant original literature is used. For example, the importance of the seed in plant miRNA-target interactions was established in the first papers on PHABULOSA repression by miR165/166 in 2003/2004 by the Martienssen and Bartel labs, and studies on other miRNA-target interactions by the Bartel lab in 2004-2006 further confirmed and generalized this notion. Similarly, the paper used to cite the introduction of the seed notion (Brennecke/Stark 2004) is certainly important and relevant, but the earlier contributions from Eric Lai (2002) and David Bartel (2002-2003) should not be omitted. Similarly, it would be relevant to include mention of the very clear biochemical proof of its importance (Ameres, Cell 2007) as well as the clear understanding of its importance from the variety of prokaryotic and eukaryotic Ago structures. I would be grateful if the authors would take considerably more care with the referencing in a revised version for this or another journal.

Answer:

We added citations from: Mallory et al, 2004; Bartel, 2004; Lai, 2002; Rhoades et al, 2002; Lewis et al, 2003; Ameres et al, 2007, according to the reviewer's request.

We also added citations from: Aukerman & Sakai, 2003; Chen, 2004; Gandikota et al, 2007; Brodersen et al, 2008.

As a very final point, I wish to thank the authors for undertaking this work that I think is very important for refining our understanding of the small RNA regulatory systems came into existence and what is important for how they work. I am, for example, quite curious to learn what the examination of the Piwi loop in cnidarian Agos gives!

Referee #3 (Comments on Novelty/Model System for Author):

As detailed in the remarks to the authors, there is a mismatch between the statistical analyses and the conclusions drawn.

Referee #3 (Remarks for Author):

The authors investigate target recognition of cnidarian miRNA mimics. They detect repression of reporters with sites that have extensive complementarity to the miRNA, but they are unable to detect significant repression of reporters with sites that match only the seed of the miRNA. Thus seed-based targeting appears to have emerged in the bilaterian clade, after the divergence of cnidarians and bilaterians. Although this

fundamental insight into miRNA evolution is of interest, the way in which the authors interpret this finding is of major concern.

Major concern

1. The results of the current paper do not provide useful information for evaluating whether plant and animal microRNAs have a common origin. The RNAi pathway (i.e., the use of siRNAs derived from Dicer processing of long dsRNA to direct AGO-catalyzed target slicing) is found broadly throughout most eukaryotic species, and thus is thought to have been present in the last eukaryotic common ancestor (LECA). By contrast, miRNAs are not found broadly throughout eukaryotes, and are instead found in only a small, polyphyletic subset of eukaryotic clades, scattered among a few of the 25 major clades the eukaryotic tree (for tree, see Burki et al., PMID: 31606140), suggesting that the extant miRNA pathways were not in the LECA but instead each emerged independently from the ancestral RNAi pathway. As expected for this independent emergence, the miRNAs in different clades have no evidence of sequence conservation. For example, the miRNAs conserved in plants have no resemblance to those conserved in animals or other eukaryotic clades. Moreover, where investigated, the miRNAs in different clades have quite different biogenesis pathways. For example, plant pri-miRNAs have very different secondary structures than animal miRNAs, and plant pri-miRNAs are processed by a single endonuclease in the nucleus, whereas animal pri-miRNAs are processed one endonuclease in the nucleus and then by a second in the cytoplasm. Thus, miRNA pathways appear to have arisen after the LECA (and after the earliest eukaryotic divergence, which separated the lineage that gave rise to plants from that that gave rise to animals; Burki, et al.). Note that the authors favor an alternative model of miRNA evolution in which miRNAs were in the common ancestor of plants and animals (i.e., in the LECA), citing the finding that miRNAs of both plants and cnidaria are often methylated and that both use the Hyl1 co-factor for Dicer processing. However, both of these findings could also be explained by the presence of methylation and Hyl1 in an ancestral RNAi pathway, and the alternative model would require miRNAs to have been independently lost in a very large number of eukaryotic lineages (see Burki et al., all curiously not shown in Figure 4)-a far less parsimonious scenario than the conventional model of independent gain in a few lineages.

In the conventional model of miRNA evolution, each new miRNA pathway evolved from the ancestral RNAi pathway with the independent emergence of a cellular encoded short RNA hairpin that could be cleaved by Dicer to produce a guide RNA that directs the AGO-catalyzed slicing of a cellular mRNA. Each nascent miRNA pathway gradually diverged from the RNAi pathway, evolving specialized Dicer and AGO proteins. The animal pathway further diverged to evolve seed-based target repression, in addition to its ancestral AGO-catalyzed target slicing, although humans and many other animals have retained AGO-catalyzed slicing of some targets. This paper explores the question of when seed-based target repression emerged in the animal lineage, concluding that seed-based repression emerged after the divergence of cnidarians and bilaterians. This conclusion is mistakenly used to support the alternative model of miRNA evolution in which miRNAs were in the common ancestor of plants and animals (i.e., in the LECA). However, in both the conventional model and this alternative model of miRNA evolution, bilaterian miRNAs are thought to have descended from a pathway in which targets underwent AGO-catalyzed slicing. Thus, the conclusion that seed-based repression emerged after the divergence of cnidarians cannot be used to distinguish between the two models,

since both models propose that the seed-based repression descended from a pathway centered on AGO-catalyzed slicing. Therefore, the title and text should be revised to remove all claims that the results support a common origin for plant and animal miRNAs.

Answer:

We would like to address the reviewer's concerns. Regarding the claim that miRNAs are found in a polyphyletic group of eukaryotes, we would like to remind that loss of the miRNA system is possible and was observed in a number of lineages such as ctenophores and placozoans (Grimson *et al*, 2008; Maxwell *et al*, 2012). The loss of the miRNA system in eukaryotic group might be advantageous and probable, as demonstrated by the loss of the RNAi system in lineages of fungi (Drinnenberg *et al*, 2011). In addition, many eukaryotic groups haven't been characterized for the presence of the miRNA system and one cannot rule out more widespread presence in the eukaryotic tree, as new information can be revealed, demonstrated by the recent identification of miRNAs in Amoebozoa and other single cell eukaryotes (Bråte *et al*, 2018; Edelbroek *et al*, 2024).

In response to the point made by the reviewer that lack of miRNA sequence conservation between plants and animals indicates independent emergence, please refer to reply to reviewer #2 point 3, where we discuss the evidence that miRNA sequence conservation is very rare even in lineages that have a common origin for the miRNA system such as sponges, plants, Cnidaria and Bilateria.

Regarding the differences in biogenesis between plants and animals, the involvement of HYL1 and HEN1 between plants and Cnidaria in the biogenesis of miRNAs might alternatively be interpreted as convergence, however, this is less parsimonious than an ancestral role of HYL1 in miRNA biogenesis and its replacement in Bilateria. Moreover, HYL1 was also reported in sponges (Moran *et al*, 2013). It is also notable that HYL1 localization in *Nematostella* cells were not tested, hence, it is possible that it operates in the nucleus similarly to plants.

In conclusion, our study follows a line of recent evidence that supports the alternative scenario of common origin of the miRNA system. Following the discovery that miRNAs in *Nematostella* act in a similar manner to plants miRNAs by fully matching and cleaving their targets, the aim of our present study was to investigate whether the bilaterian seed match is a possible mode of action in cnidarian miRNAs. Our results show that this isn't the case and therefore support the notion that full complementarity between miRNAs and their targets is conserved between plants and cnidarians. Hence, as mentioned before, the current results join previous findings that support a common origin of the miRNA system. While we strongly believe in the validity of this homology scenario, we do acknowledge that there is always a risk in interpreting extant states in order to understand ancient ones and hence we did try to convey a more cautious tone in some points in the text and we do address in the discussion the alternative scenario based on convergence.

2. The statistical analyses do not support the primary biological claim of the paper. The paper claims that cnidarians do not perform seed-based target recognition. This conclusion is based on not seeing a statistically significant repression of targets with seed pairing. However, not being confident in a difference is not the same as being confident in no difference, especially when data are noisy. To conclude that seed pairing does not mediate repression, the authors should perform a statistical analysis

that reports their confidence that seed pairing causes no repression. This is particularly important for the *Hydractinia* results.

Answer:

We would first like to note that the *Hydractinia* experiments were modified. We added a fluorescent tracer for normalization of injected mRNA expression and measured fluorescence intensity instead of protein concentration. Please see also our response to Referee #1 and Figure 3. As the reviewer suggested, we added a statistical test for equivalence of means. Following consultation with an expert on biostatistics (Prof. Liran Carmel, Department of Genetics, The Hebrew University of Jerusalem) we used Welch Modified Two-Sample t-Test (TOST) to test the equivalence of normalized GFP intensity of the negative control and the seed groups. The test concluded that the bounds of a 90% confidence interval fall between the equivalence bounds of -0.19 and 0.19. The null equivalence hypothesis is rejected with p-value of 0.047 and the null significance hypothesis that the effect is equal to zero is not significant. When testing the means of the negative control and the positive control groups, the test is significant with bounds of -0.36 and 0.36 (but not between -0.35 and 0.35), and the null equivalence hypothesis is rejected with p-value of 0.044. Under these selected bounds the null significance hypothesis that the effect is equal to zero is rejected. The results of the equivalence tests show that the equivalence bounds within them the means of the negative control and seed groups are equivalents are almost twice smaller than the bounds defined for means of negative control and positive control groups. We believe that these results combined with the modified *Hydractinia* experiments support that the seed match mimiR did not cause a knockdown of GFP mRNA with three seed binding sites in *Hydractinia*.

22nd Aug 2024

Dear Yehu,

Thank you for the submission of your revised manuscript. I had already informed you about the expected delay in the review process but we have now received the reports from both referees that were asked to assess it. As you will see, the referees acknowledge that you have addressed most of their concerns and added new data that strengthened your study. However, you will also see that both referees continue to raise concerns regarding the evolutionary implications of your results and the proposed common evolutionary origin of plant and animal miRNAs.

Since both referees raise similar concerns and consider your data insufficient to conclusively rule out the conventional model of miRNA evolution, I propose to follow the referee's advice to discuss your data in a balanced way without sharply concluding on a common evolutionary origin, as referee 2 phrased it.

In addition to these textual changes, I noticed a few editorial points that need to be addressed, as follows:

- Your manuscript will be published in our Reports section. To comply with this format, we kindly ask you to combine the Results and Discussion section.
- Please enter all information on funding in the online manuscript tracking system. The information in the system must match that in the manuscript.
- Please rename the Competing Interests statement to "Disclosure Statement and Competing Interests"
- Figure callouts should be in numerical and alphabetical order. In this context we note that Fig. 3A-B is called out before Fig. 2D-F. If possible, please modify these figures.
- Author Checklist: Ethics/studies involving experimental animals. You state that the authority granting approval is specified in the Materials and Methods section. I could however not find this information. Can you please clarify?
- Since July this year we have mandated the Structured Methods format. To comply with this format, could you please add a Reagents and Tools table? You can download the template (.docx) using this link:
: <https://www.embopress.org/page/journal/14693178/authorguide#structuredmethods>.

The file should be uploaded as separate file with the file type "Reagent Table". If you prefer to keep the template sequences in Table 2, you can refer to that table in the R&T table instead of inserting the sequences.

- Tables: please add the name, e.g., "Table 1", on top of the table and the legend is underneath the table. E.g., Table 1 - Pairwise comparisons - followed by the table - followed by the legend.
- The correct titles for the EV figure legends in the manuscript should be: Figure EV1, etc. instead of Expanded view Figure1, etc.
- Please provide the specific URL for the PRJNA1111011 dataset in the data availability statement.
- Our production/data editors have asked you to clarify several points in the figure legends (see below). Please incorporate these changes in the manuscript and return the revised file with tracked changes with your final manuscript submission.

A) Statistical test information. Only p-values that are actually shown in the figure panel(s) should (and must) be defined in the legends, all others should be removed from (or added to) the legend. Moreover, we ask for the specification of exact p-values:

1. Please define the annotated p values * as well as provide the exact p-values for the same in the legend of figure EV 3; as appropriate.
2. Please note that the exact p values are not provided in the legends of figures 1g-h; 2c; 3b; 4b-c.
3. Please note that in figures 1g-h; 2c; 3b; 4b-c; there is a mismatch between the annotated p values in the figure legend and the annotated p values in the figure file that should be corrected.

B) Replicates and error bars:

4. Please note that the box plot needs to be defined in terms of minima, maxima, centre, bounds of box and whiskers, and percentile in the legend of figure EV 3.
5. Please note that the box plots need to be defined in terms of bounds of box and percentile in the legends of figures 1g-h; 2c; 3b; 4b-c.

- You state in the methods: "Crystal structures of atAGO10 (<https://doi.org/10.2210/pdb7SVA/pdb>) and hsAGO2 (<https://doi.org/10.2210/pdb4OLA/pdb>) were taken from PDB database (Schirle & MacRae, 2012; Xiao et al, 2023b)".

You could use the Data Citation format in this case, where you cite first the paper reporting the structures and in addition the structural dataset itself as a Data reference.

In the main text, data citations are formatted as follows: "Data ref: Smith et al, 2001" or "Data ref: NCBI Sequence Read Archive PRJNA342805, 2017". In the Reference list, data citations must be labeled with "[DATASET]". A data reference must provide the database name, accession number/identifiers and a resolvable link to the landing page from which the data can be accessed at the end of the reference. Further instructions are available at .

- The Source Data for the main figures need to be uploaded as one folder per figure.

- We perform a routine check on all .xls source data files submitted. In this case I noticed a potential duplication in the source data for Figure 1G. Can you please check the color-coded attached .xls file? It might be a copy-and-paste error.

- Finally, EMBO Reports papers are accompanied online by

A) a short (1-2 sentences) summary of the findings and their significance,

B) 2-3 bullet points highlighting key results and

C) a schematic summary figure that provides a sketch of the major findings (not a data image).

Please provide the summary figure as a separate file in PNG or JPG format at a size of 550x300-600 pixels (width x height).

Please note that the size is rather small and that text needs to be readable at the final size. Please send us this information along with the revised manuscript.

- On a different note, I would like to alert you that EMBO Press offers a new format for a video-synopsis of work published with us, which essentially is a short, author-generated film explaining the core findings in hand drawings, and, as we believe, can be very useful to increase visibility of the work. This has proven to offer a nice opportunity for exposure i.p. for the first author(s) of the study. Please see the following link for representative examples and their integration into the article web page:

<https://www.embopress.org/doi/full/10.15252/embj.2019103932>

With kind regards,

Martina

*** Instructions to submit your revised manuscript ***

*** PLEASE NOTE *** As part of the EMBO Publications transparent editorial process initiative (see our Editorial at <https://www.embopress.org/doi/pdf/10.1002/emmm.201000094>), EMBO will publish online a Review Process File to accompany accepted manuscripts.

In the event of acceptance, this file will be published in conjunction with your paper and will include the anonymous referee reports, your point-by-point response and all pertinent correspondence relating to the manuscript. If you do NOT want this file to be published, please inform the editorial office.

1) a .docx formatted version of the manuscript text (including Figure legends and tables)

2) Separate figure files*

3) supplemental information as Expanded View and/or Appendix. Please carefully check the authors guidelines for formatting Expanded view and Appendix figures and tables at <https://www.embopress.org/page/journal/17574684/authorguide#expandedview>

4) a letter INCLUDING the reviewer's reports and your detailed responses to their comments (as Word file).

5) The paper explained: EMBO Molecular Medicine articles are accompanied by a summary of the articles to emphasize the major findings in the paper and their medical implications for the non-specialist reader. Please provide a draft summary of your article highlighting

6) For more information: There is space at the end of each article to list relevant web links for further consultation by our readers. Could you identify some relevant ones and provide such information as well? Some examples are patient associations, relevant databases, OMIM/proteins/genes links, author's websites, etc...

7) Author contributions: the contribution of every author must be detailed in a separate section.

8) EMBO Press now requires a complete author checklist (<https://www.embopress.org/page/journal/17574684/authorguide>) to be submitted with all revised manuscripts. Please use the checklist as guideline for the sort of information we need WITHIN the manuscript. The checklist should only be filled with page numbers where the information can be found. This is particularly important for animal reporting, antibody dilutions (missing) and exact values and n that should be indicated instead of a range.

9) Every published paper now includes a 'Synopsis' to further enhance discoverability. Synopses are displayed on the journal webpage and are freely accessible to all readers. They include a short stand first (maximum of 300 characters, including space) as well as 2-5 one sentence bullet points that summarise the paper. Please write the bullet points to summarise the key NEW findings. They should be designed to be complementary to the abstract - i.e. not repeat the same text. We encourage inclusion of key acronyms and quantitative information (maximum of 30 words / bullet point). Please use the passive voice. Please attach these in a separate file or send them by email, we will incorporate them accordingly.

You are also welcome to suggest a striking image or visual abstract to illustrate your article. If you do please provide a jpeg file 550 px-wide x 300-600px high.

10) A Conflict of Interest statement should be provided in the main text

11) Please note that we now mandate that all corresponding authors list an ORCID digital identifier. This takes <90 seconds to complete. We encourage all authors to supply an ORCID identifier, which will be linked to their name for unambiguous name identification.

Currently, our records indicate that the ORCID for your account is 0000-0001-9928-9294.

Link Not Available

12) Include a Reagents and Tools Table as part of the Methods section, which can be downloaded from our author guidelines (<https://www.embopress.org/page/journal/17574684/authorguide#structuredmethods>)

Photos 400-800 DPI

*Additional important information regarding figures and illustrations can be found at <https://bit.ly/EMBOPressFigurePreparationGuideline>. See also figure legend preparation guidelines: <https://www.embopress.org/page/journal/17574684/authorguide#figureformat>

**** Reviewer's comments ****

Referee #2 (Comments on Novelty/Model System for Author):

I am not sure this is really relevant, because as I understand it, the review process is being run by EMM even if the manuscript is actually submitted to EMBO Reports?

Referee #2 (Remarks for Author):

In their revised manuscript, the authors have added valuable experimental results that strengthen the paper considerably. I find the paper very interesting, but I still believe that the authors over-conclude on the evolutionary implications of their results. I agree that the fact that no common miRNA exists between animals and plants does not definitively prove that miRNAs evolved independently in them, but I would argue that it remains a very likely possibility. Conversely, the results presented here do not prove that miRNAs must have a common evolutionary origin in animals of plants. Please note also that it is not completely true to state that miRNAs as such are evolutionarily fluid. At least 7 miRNAs (miR156, miR159, miR160, miR166, miR171, miR390, miR408) are conserved across ~500M years of land plant evolution from liverworts to angiosperms.

I recommend that the authors follow the advice of the editor in the remarks on the originally submitted version. The work is really interesting and has a lot of merit without necessarily trying to conclude sharply that there has to be a common evolutionary origin because miRNAs guide slicing in cnidarians. If the authors modify title and abstract accordingly, and deliver a balanced discussion that discusses the possibility that miRNAs may have a common evolutionary origin based on their results, but also lists arguments against such an interpretation, I would clearly support its publication.

Referee #3 (Remarks for Author):

The authors have addressed my second concern, which regarded the statistics. However, they have not satisfactorily addressed my first major concern (repeated in the next two paragraphs), which regarded the evolutionary implications of their results:

"Major concern 1. The results of the current paper do not provide useful information for evaluating whether plant and animal microRNAs have a common origin. The RNAi pathway (i.e., the use of siRNAs derived from Dicer processing of long dsRNA to direct AGO-catalyzed target slicing) is found broadly throughout most eukaryotic species, and thus is thought to have been present in the last eukaryotic common ancestor (LECA). By contrast, miRNAs are not found broadly throughout eukaryotes, and are instead found in only a small, polyphyletic subset of eukaryotic clades, scattered among a few of the 25 major clades the eukaryotic tree (for tree, see Burki et al., PMID: 31606140), suggesting that the extant miRNA pathways were not in the LECA but instead each emerged independently from the ancestral RNAi pathway. As expected for this independent emergence, the miRNAs in different clades have no evidence of sequence conservation. For example, the miRNAs conserved in plants have no resemblance to those conserved in animals or other eukaryotic clades. Moreover, where investigated, the miRNAs in different clades have quite different biogenesis pathways. For example, plant pri-miRNAs have very different secondary structures than animal miRNAs, and plant pri-miRNAs are processed by a single endonuclease in the nucleus, whereas animal pri-miRNAs are processed one endonuclease in the nucleus and then by a second in the cytoplasm. Thus, miRNA pathways appear to have arisen after the LECA (and after the earliest eukaryotic divergence, which separated the lineage that gave rise to plants from that that gave rise to animals; Burki, et al.). Note that the authors favor an alternative model of miRNA evolution in which miRNAs were in the common ancestor of plants and animals (i.e., in the LECA), citing the finding that miRNAs of both plants and cnidaria are often methylated and that both use the Hyl1 co-factor for Dicer processing. However, both of these findings could also be explained by the presence of methylation and Hyl1 in an ancestral RNAi pathway, and the alternative model would require miRNAs to have been independently lost in a very large number of eukaryotic lineages (see Burki et al., all curiously not shown in Figure 4)-a far less parsimonious scenario than the conventional model of independent gain in a few lineages.

In the conventional model of miRNA evolution, each new miRNA pathway evolved from the ancestral RNAi pathway with the independent emergence of a cellular encoded short RNA hairpin that could be cleaved by Dicer to produce a guide RNA that directs the AGO-catalyzed slicing of a cellular mRNA. Each nascent miRNA pathway gradually diverged from the RNAi pathway, evolving specialized Dicer and AGO proteins. The animal pathway further diverged to evolve seed-based target repression, in addition to its ancestral AGO-catalyzed target slicing, although humans and many other animals have retained AGO-catalyzed slicing of some targets. This paper explores the question of when seed-based target repression emerged in the animal lineage, concluding that seed-based repression emerged after the divergence of cnidarians and bilaterians. This conclusion is mistakenly

used to support the alternative model of miRNA evolution in which miRNAs were in the common ancestor of plants and animals (i.e., in the LECA). However, in both the conventional model and this alternative model of miRNA evolution, bilaterian miRNAs are thought to have descended from a pathway in which targets underwent AGO-catalyzed slicing. Thus, the conclusion that seed-based repression emerged after the divergence of cnidarians cannot be used to distinguish between the two models, since both models propose that the seed-based repression descended from a pathway centered on AGO-catalyzed slicing. Therefore, the title and text should be revised to remove all claims that the results support a common origin for plant and animal miRNAs."

1) In their rebuttal, the authors responded by saying that a couple animal lineages have lost miRNAs and some fungal lineages have lost the entire RNAi pathway, implying that widespread loss among the 25 different eukaryotic lineages might be possible. They also bring up the possibility that the presence of miRNAs in these different lineages might be vastly underappreciated. In their paper, the authors completely ignore my concern. They also continue to promote an extremely simplistic view of eukaryotic evolution (Figure 5), which might have been acceptable decades ago, when as "crown species" plants and animals were thought to be more related to each other than to other eukaryotes, but we now know that this view is no longer valid. The authors should update their evolutionary model and its implications to be consistent with the current understanding of eukaryotic evolution, as follows:

- a) Clearly state that if the animal and plant miRNAs pathways share a common origin, then the LECA would have had these ancestral miRNAs.
- b) Clearly state that their proposal that the LECA had miRNAs implies that either miRNAs are much more prevalent in eukaryotic lineages than currently appreciated or there has been incredibly frequent loss of miRNAs in most of the 25 major eukaryotic lineages.
- c) Clearly state that the scenario of widespread loss of miRNAs in most of the 25 major lineages is less parsimonious than the conventional model of independent gain of miRNAs in a few lineages.
- d) Figure 5 should show the branching of the other 23 major eukaryotic lineages and indicate the microRNA status of each lineage, even if it is unknown.

2) In their rebuttal and revised manuscript, the authors do not address my point regarding the fact that their new results do not speak to the question of common origins of plant and animal miRNAs. The conclusion that seed-based repression emerged after the divergence of cnidarians cannot be used to distinguish between the conventional and alternative models of plant and animal miRNA origins, since both models propose that the seed-based repression descended from a pathway centered on AGO-catalyzed slicing. In the conventional model, plant and animal miRNAs emerged independently, each from the ancestral RNAi pathway centered on AGO-catalyzed slicing. Thus, the idea that animal miRNAs initially acted through slicing and seed-based targeting only emerged after the split with cnidaria is totally consistent with the conventional model. Therefore, the title, abstract, and text should be revised to remove all claims that the new results suggest a common origin for plant and animal miRNAs, and the text should clearly state that the new results of this paper are fully consistent with either a common origin of plant and animal miRNAs or an independent origin of plant and animal miRNAs.

Editor notes:

As you will see, the referees acknowledge that you have addressed most of their concerns and added new data that strengthened your study. However, you will also see that both referees continue to raise concerns regarding the evolutionary implications of your results and the proposed common evolutionary origin of plant and animal miRNAs.

Since both referees raise similar concerns and consider your data insufficient to conclusively rule out the conventional model of miRNA evolution, I propose to follow the referee's advice to discuss your data in a balanced way without sharply concluding on a common evolutionary origin, as referee 2 phrased it.

**** Reviewer's comments ****

Referee #2 (Comments on Novelty/Model System for Author):

I am not sure this is really relevant, because as I understand it, the review process is being run by EMM even if the manuscript is actually submitted to EMBO Reports?

Answer:

This is done purely for technical reasons according to the request of the editorial team.

Referee #2 (Remarks for Author):

In their revised manuscript, the authors have added valuable experimental results that strengthen the paper considerably.

I find the paper very interesting, but I still believe that the authors over-conclude on the evolutionary implications of their results. I agree that the fact that no common miRNA exists between animals and plants does not definitively prove that miRNAs evolved independently in them, but I would argue that it remains a very likely possibility. Conversely, the results presented here do not prove that miRNAs must have a common evolutionary origin in animals or plants. Please note also that it is not completely true to state that miRNAs as such are evolutionarily fluid. At least 7 miRNAs (miR156, miR159, miR160, miR166, miR171, miR390, miR408) are conserved across ~500M years of land plant evolution from liverworts to angiosperms.

I recommend that the authors follow the advice of the editor in the remarks on the originally submitted version. The work is really interesting and has a lot of merit without necessarily trying to conclude sharply that there has to be a common evolutionary origin because miRNAs guide slicing in cnidarians. If the authors modify title and abstract accordingly, and deliver a balanced discussion that discusses the possibility that miRNAs may have a common evolutionary origin based on their results, but also lists arguments against such an interpretation, I would clearly support its publication.

Answer:

We thank the referee for their helpful remarks. We have edited the text to include a more balanced discussion regarding the two possible scenarios of miRNA evolution. We have also edited the title of the manuscript accordingly. We would like to note that the current version of the abstract already mentions a common evolutionary origin as a possibility and not a conclusion, thus we have not made such changes to the abstract.

Referee #3 (Remarks for Author):

The authors have addressed my second concern, which regarded the statistics. However, they have not satisfactorily addressed my first major concern (repeated in the next two paragraphs), which regarded the evolutionary implications of their results:

"Major concern 1. The results of the current paper do not provide useful information for evaluating whether plant and animal microRNAs have a common origin. The RNAi pathway (i.e., the use of siRNAs derived from Dicer processing of long dsRNA to direct AGO-catalyzed target slicing) is found broadly throughout most eukaryotic species, and thus is thought to have been present in the last eukaryotic common ancestor (LECA). By contrast, miRNAs are not found broadly throughout eukaryotes, and are instead found in only a small, polyphyletic subset of eukaryotic clades, scattered among a few of the 25 major clades the eukaryotic tree (for tree, see Burki et al., PMID: 31606140), suggesting that the extant miRNA pathways were not in the LECA but instead each emerged independently from the ancestral RNAi pathway. As expected for this independent emergence, the miRNAs in different clades have no evidence of sequence conservation. For example, the miRNAs conserved in plants have no resemblance to those conserved in animals or other eukaryotic clades. Moreover, where investigated, the miRNAs in different clades have quite different biogenesis pathways. For example, plant pri-miRNAs have very different secondary structures than animal miRNAs, and plant pri-miRNAs are processed by a single endonuclease in the nucleus, whereas animal pri-miRNAs are processed one endonuclease in the nucleus and then by a second in the cytoplasm. Thus, miRNA pathways appear to have arisen after the LECA (and after the earliest eukaryotic divergence, which separated the lineage that gave rise to plants from that that gave rise to animals; Burki, et al.). Note that the authors favor an alternative model of miRNA evolution in which miRNAs were in the common ancestor of plants and animals (i.e., in the LECA), citing the finding that miRNAs of both plants and cnidaria are often methylated and that both use the Hyl1 co-factor for Dicer processing. However, both of these findings could also be explained by the presence of methylation and Hyl1 in an ancestral RNAi pathway, and the alternative model would require miRNAs to have been independently lost in a very large number of eukaryotic lineages (see Burki et al., all curiously not shown in Figure 4)-a far less parsimonious scenario than the conventional model of independent gain in a few lineages.

In the conventional model of miRNA evolution, each new miRNA pathway evolved from the ancestral RNAi pathway with the independent emergence of a cellular encoded short RNA hairpin that could be cleaved by Dicer to produce a guide RNA that directs the AGO-catalyzed slicing of a cellular mRNA. Each nascent miRNA pathway gradually diverged from the RNAi pathway, evolving specialized Dicer and AGO proteins. The animal pathway further diverged to evolve seed-based target repression, in addition to its ancestral AGO-catalyzed target slicing, although humans

and many other animals have retained AGO-catalyzed slicing of some targets. This paper explores the question of when seed-based target repression emerged in the animal lineage, concluding that seed-based repression emerged after the divergence of cnidarians and bilaterians. This conclusion is mistakenly used to support the alternative model of miRNA evolution in which miRNAs were in the common ancestor of plants and animals (i.e., in the LECA). However, in both the conventional model and this alternative model of miRNA evolution, bilaterian miRNAs are thought to have descended from a pathway in which targets underwent AGO-catalyzed slicing. Thus, the conclusion that seed-based repression emerged after the divergence of cnidarians cannot be used to distinguish between the two models, since both models propose that the seed-based repression descended from a pathway centered on AGO-catalyzed slicing. Therefore, the title and text should be revised to remove all claims that the results support a common origin for plant and animal miRNAs."

1) In their rebuttal, the authors responded by saying that a couple animal lineages have lost miRNAs and some fungal lineages have lost the entire RNAi pathway, implying that widespread loss among the 25 different eukaryotic lineages might be possible. They also bring up the possibility that the presence of miRNAs in these different lineages might be vastly underappreciated. In their paper, the authors completely ignore my concern. They also continue to promote an extremely simplistic view of eukaryotic evolution (Figure 5), which might have been acceptable decades ago, when as "crown species" plants and animals were thought to be more related to each other than to other eukaryotes, but we now know that this view is no longer valid. The authors should update their evolutionary model and its implications to be consistent with the current understanding of eukaryotic evolution, as follows:
a) Clearly state that if the animal and plant miRNAs pathways share a common origin, then the LECA would have had these ancestral miRNAs.

Answer:

We have added a statement to the discussion as the referee requested, stating that if plant and animal miRNAs share an ancestral origin, then some ancestral miRNAs could have existed in their last common ancestor and that the lack of shared miRNAs can support a scenario of convergence. Please see in addition our reply to reviewer 2.

b) Clearly state that their proposal that the LECA had miRNAs implies that either miRNAs are much more prevalent in eukaryotic lineages than currently appreciated or there has been incredibly frequent loss of miRNAs in most of the 25 major eukaryotic lineages.

Answer:

We have added statements to the discussion regarding the possibility that miRNAs exist in more lineages than currently known, and that the miRNA system could have been lost multiple times, as most eukaryotic lineages are not known to possess miRNAs. Yet, we also state the fact that many eukaryotic lineages are very much understudied when it comes to small RNA sequencing.

c) Clearly state that the scenario of widespread loss of miRNAs in most of the 25

major lineages is less parsimonious than the conventional model of independent gain of miRNAs in a few lineages.

Answer:

We thank the referee for this comment. We would like to note that parsimony is indeed an important principle in evolution, however, there are additional principles that are found to shape evolutionary processes, and cases when the number of changes is found not to be minimal.

d) Figure 5 should show the branching of the other 23 major eukaryotic lineages and indicate the microRNA status of each lineage, even if it is unknown.

Answer:

We thank the referee for the comment. We believe that adding 23 lineages to the Figure 6 (originally Figure 5) will make it unclear. To comply with the reviewer's suggestion, we have edited the title of the figure to emphasize that it shows a simplified tree highlighting similarities and differences in the miRNA system in plants, Cnidaria and Bilateria.

2) In their rebuttal and revised manuscript, the authors do not address my point regarding the fact that their new results do not speak to the question of common origins of plant and animal miRNAs. The conclusion that seed-based repression emerged after the divergence of cnidarians cannot be used to distinguish between the conventional and alternative models of plant and animal miRNA origins, since both models propose that the seed-based repression descended from a pathway centered on AGO-catalyzed slicing. In the conventional model, plant and animal miRNAs emerged independently, each from the ancestral RNAi pathway centered on AGO-catalyzed slicing. Thus, the idea that animal miRNAs initially acted through slicing and seed-based targeting only emerged after the split with cnidaria is totally consistent with the conventional model. Therefore, the title, abstract, and text should be revised to remove all claims that the new results suggest a common origin for plant and animal miRNAs, and the text should clearly state that the new results of this paper are fully consistent with either a common origin of plant and animal miRNAs or an independent origin of plant and animal miRNAs.

Answer:

Please see our above reply to Referee 2.

18th Nov 2024

Dear Yehu,

Thank you for submitting your revised manuscript and please accept my apologies for the delay in handling it. I have now looked at everything and all is fine. Therefore, I am very are pleased to accept your manuscript for publication in EMBO Reports.

Kind regards,

Martina
